# On the influence of prior information evaluated by fully Bayesian criteria in a personalized whole-brain model of epilepsy spread

**Meysam Hashemi** [1]*, **Anirudh N. Vattikonda** [1], **Viktor Sip** [1], **Sandra Diaz-Pier** [2], **Alexander Peyser** [2,3], **Huifang Wang** [1], **Maxime Guye** [4], **Fabrice Bartolomei** [5], **Marmaduke M. Woodman** [1], **Viktor K. Jirsa** [1]*

**1** Aix Marseille Univ, INSERM, INS, Inst Neurosci Syst, Marseille, France, **2** SimLab Neuroscience, Jülich Supercomputing Centre (JSC), Institute for Advanced Simulation, JARA, Forschungszentrum Jülich GmbH, Jülich, Germany, **3** Google, München, Germany, **4** Aix Marseille Univ, CNRS, CRMBM, Marseille, France, **5** Epileptology Department, and Clinical Neurophysiology Department, Assistance Publique des Hôpitaux de Marseille, Marseille, France

* meysam.hashemi@univ-amu.fr (MH); viktor.jirsa@univ-amu.fr (VKJ)

**Data Availability Statement:** The patient data sets cannot be made publicly available due to the data protection concerns regarding potentially

## Abstract

Individualized anatomical information has been used as prior knowledge in Bayesian inference paradigms of whole-brain network models. However, the actual sensitivity to such personalized information in priors is still unknown. In this study, we introduce the use of fully Bayesian information criteria and leave-one-out cross-validation technique on the subject-specific information to assess different epileptogenicity hypotheses regarding the location of pathological brain areas based on a priori knowledge from dynamical system properties. The Bayesian Virtual Epileptic Patient (BVEP) model, which relies on the fusion of structural data of individuals, a generative model of epileptiform discharges, and a self-tuning Monte Carlo sampling algorithm, is used to infer the spatial map of epileptogenicity across different brain areas. Our results indicate that measuring the out-of-sample prediction accuracy of the BVEP model with informative priors enables reliable and efficient evaluation of potential hypotheses regarding the degree of epileptogenicity across different brain regions. In contrast, while using uninformative priors, the information criteria are unable to provide strong evidence about the epileptogenicity of brain areas. We also show that the fully Bayesian criteria correctly assess different hypotheses about both structural and functional components of whole-brain models that differ across individuals. The fully Bayesian information-theory based approach used in this study suggests a patient-specific strategy for epileptogenicity hypothesis testing in generative brain network models of epilepsy to improve surgical outcomes.

identifying and sensitive patient information. Interested researchers may access the data sets by contacting Clinical Data Manager Aurélie Ponz (aurelie.ponz@univ-amu.fr) at the Institut de Neurosciences des Systèmes, Aix-Marseille Université. The main source codes needed to reproduce the presented results are available on GitHub (https://github.com/ins-amu/BVEP).

**Funding:** This work was funded by the French National Research Agency (ANR) as part of the second "Investissements d'Avenir" program, ANR-17-RHUS-0004, EPINOV (https://anr.fr) to VJ, FB, and MG, by European Union's Horizon 2020 Framework Programme for Research and Innovation under the Specific Grant Agreement No. 785907 and 945539, Human Brain Project SGA2 and SGA3 (https://ec.europa.eu/programmes/horizon2020) to VJ, by European Union's Horizon 2020 Framework Programme for Research and Innovation under the Specific Grant Agreement and No. 826421, VirtualBrainCloud (https://ec.europa.eu/programmes/horizon2020) to VJ, PHRC-I 2013 EPISODIUM (grant number 2014-27), the Fondation pour la Recherche Médicale (DIC20161236442), and by SATT Sud-Est, 827-SA-16-UAM (https://www.sattse.com) to VJ, FB, and MG. The funders had no role in study design, data collection and analysis, decision to publish, or preparation of the manuscript.

**Competing interests:** The authors have declared that no competing interests exist.

**Abbreviations:** AIC, Akaike's information criterion; BIC, Bayesian information criterion; BVEP, Bayesian virtual epileptic patient; CV, cross-validation; EZ, epileptogenic zone; HMC, Hamiltonian Monte Carlo; HPC, High Performance Computing; HZ, healthy zone; LOO, leave-one-out cross-validation; MCMC, Markov Chain Monte Carlo; MLE, maximum likelihood estimate; NUTS, No-U-Turn Sampler; PPL, probabilistic programming language; PZ, propagation zone; SC, structural connectivity; TVB, The Virtual Brain; VEP, virtual epileptic patient; WAIC, Watanabe-Akaike information criterion.

## Author summary

Reliable prediction of the Epileptogenic Zone (EZ) is a challenging task due to nontrivial brain network effects, non-linearity involved in spatiotemporal brain organization, and uncertainty in prior information. Based on the whole-brain modeling approach, the anatomical information of patients can be merged with a generative model of epileptiform discharges to build a personalized large-scale brain model of epilepsy spread. Here, we apply information criteria and cross-validation technique to a whole-brain model of epilepsy spread to infer and validate the spatial map of epileptogenicity across different brain areas. By definition, classical information criteria are independent of prior information, in which the penalty term (number of parameters and observed data) is the same across different EZ candidates, making them infeasible to determine the best among a set of epileptogenicity hypotheses. In contrast, the fully Bayesian information criteria and cross-validation enable us to integrate our prior information to improve out-of-sample prediction accuracy for EZ identification. Using the dynamical system properties of a whole-brain model of epilepsy spread, and dependent on the level of prior information, the proposed approach provides accurate and reliable estimation about the degree of epileptogenicity across different brain areas. Our fully Bayesian approach relying on automatic inference suggests a patient-specific strategy for EZ prediction and hypothesis testing before therapeutic interventions.

## Introduction

Mathematical models have long played a key role in understanding the biological systems. However, the intrinsic uncertainty associated with parameters of dynamical models translates into uncertainty in model predictions, thereby, in the model selection among a set of candidates having best balance between complexity and accuracy. In practice, the nonlinearity and complexity of most dynamical models render the model calibration and hypothesis testing nontrivial [1–4]. Bayesian framework offers a powerful approach to deal with model uncertainty in parameter estimation, and a principled method for model prediction with a broad range of applications [5–13]. Bayesian inference techniques provide the posterior distribution of unknown parameters of the underlying data generating process given only observed responses and prior information about the underlying generative process [14, 15]. In other words, the posterior values define distributions over plausible parameter ranges consistent with observed data by updating information from both the prior and the observations. Thus, the estimated posterior is dependent on the level of information in the prior distribution, as well as the conditional probability of the observed data given the model parameters in the form of likelihood function. Markov Chain Monte Carlo (MCMC) methods are able to sample from and hence, approximate the exact posterior densities [14, 16]. However, MCMC-based sampling of posterior distributions, which converge to the desired target distribution for large enough sample size, is a challenging task due to the large curvature in the typical set [17, 18] or non-linear correlations between parameters in high-dimensional parameter spaces [19–21].

Fitting multiple models to non-invasive recording data such as the resting-state functional magnetic resonance imaging (fMRI) or electroencephalography (EEG), and then finding the best model among the set of model candidates has been established as a standard approach in neuroimaging literature [5, 22, 23]. Although the classical information criteria have been widely used in this approach [24, 25], such techniques have only recently been applied in the context of seizure prediction in order to assess different hypotheses regarding the location of

Epileptogenic Zone (EZ). The success rate of epilepsy surgery averages only about 50–60% and has not progressed in 30 years [26, 27]. Successful outcomes of surgical interventions depend critically on the ability to accurately identify the EZ in the brain, and the subsequent propagation to other close or distant brain regions. In this study, the EZ is defined as the pathological area(s) in the brain responsible for the origin and early organization of the epileptic activity [28]. Following previous studies, the term epileptogenic zone is used rather than the term epileptic focus to better describe the complexity and often spatially distributed nature of seizure initiation in the brain, not restricted to a single focus [29–31].

Large-scale brain network models emphasize the whole network character of the changes in the organization of brain activity as observed in brain disorders, which provide a useful link between brain dynamics in EZ (where the seizure starts) and the seizure propagation recruiting secondary subnetworks, the so-called Propagation Zone (PZ). While the location of the seizure onset area is usually invariant across seizures in an individual patient [31, 32], the PZ may be misclassified as the origin of pathological spatiotemporal patterns in pre-surgical monitoring. In the context of personalized brain modeling, the EZ dynamics are governed by a set of nonlinear equations to capture both seizure onset and offset, whereas the propagation depends on the interplay between node dynamics (excitability) and network coupling (structure) [4, 13]. Combining mean-field models of seizure dynamics with patient connectome describes various spatiotemporal epileptic patterns as observed in the experimental recordings [28, 33, 34]. Then, the EZ/PZ prediction by the parametric methods such as a grid-based exploration of parameter space or Approximate Bayesian Computation (ABC)-related methods [35, 36] strongly relies on the distance measure used for model evaluation. These traditional approaches suffer from the curse of dimensionality and their in-sample prediction based on low dimensional summary statistic depends critically on the tolerance level in the accepted/rejected parameter setting [37]. In contrast, the nonparametric Bayesian approach provides a robust and reliable probabilistic framework for out-of-sample prediction. Moreover, it allows us to integrate patient-specific information such as brain connectivity as the prior knowledge as well as the uncertainty regarding the organization of EZ/PZ in the model likelihood function to improve out-of-sample prediction accuracy. Although an efficient Bayesian inversion requires painstaking model-specific tuning, the probabilistic programming paradigm [38–42] featured by the self-tuning sampling algorithms [19] and automatic differentiation [43, 44] provides high-level tools to solve complex inverse problems conditioned on large data set as the observation. Despite the theoretical and technological advances, practical implications of nonparametric fully Bayesian inference with information-theoretic measures for model selection to formulate and evaluate the clinical hypothesis on the location of EZ remain to be explored.

In a Bayesian framework, for the purposes of model comparison, selection and averaging, the out-of-sample model predictive accuracy (i.e., the measure of model's ability in new data prediction) is needed following the fitting of a probabilistic model to the observed data [45–47]. For instance, in the context of modeling human brain networks, different types of mathematical models can be fitted to the neuroimaging data of subjects and then compared by out-of-sample prediction accuracy to determine the best performing among a set of compatible models [48]. This procedure can also be performed for hypothesis evaluation to explore the sensitivity and specificity of the potential hypotheses in predictions based on new data.

Several information-based criteria have been proposed for measuring the model predictive accuracy [48], however, the fully Bayesian criteria which are based on the whole posterior distribution rather than a point estimate [45, 49], have received little attention in this context. The two most popular information criteria in model selection, the Akaike information criterion (AIC [50]) and the Bayesian information criterion (BIC [51]), have been widely used to

evaluate the best balance between model complexity and accuracy [15, 48, 52], and then, were proposed as a selection rule to determine the model in the set of competitors that provides the highest amount of evidence [1, 53]. These classical information criteria are based on point estimates, which are computed independently of the prior knowledge (essentially by assuming uniform priors), while the model prediction is penalized only by a simple function of the number of data points and/or the number of model parameters [45]. Cross-validation [54–56] is another rigorous approach for estimating pointwise out-of-sample prediction accuracy. However, while using cross-validation methods to measure the model accuracy, the computational cost associated with model evaluation in training and test sets can be prohibitive for large data sets as are commonly encountered in the brain network modeling approach.

Approximate leave-one-out (LOO [57]) cross-validation and fully Bayesian information criteria such as Watanabe-Akaike (or widely applicable) information criterion (WAIC [58]) are two accurate and reliable approaches for estimating pointwise out-of-sample prediction accuracy [45, 49]. Taking the existing draws from log-likelihood evaluated at the posterior of parameter values, these approaches are able to efficiently estimate the model prediction accuracy within a negligible computational time as compared to the cost of model fitting [55, 56]. More importantly, as these methods are fully Bayesian, they enable us to integrate our prior information about the model parameters to better improve the model's ability in new data prediction [45].

The Virtual Brain (TVB) is a computational framework to simulate large-scale brain network models based on individual subject data [59, 60]. TVB is designed to simulate collective whole-brain dynamics by virtualizing brain structure and function, allowing simultaneous outputs of a number of experimental modalities. This open-source neuroinformatics platform has been extensively used to simulate common neuroimaging signals including functional MRI (fMRI), EEG, SEEG and MEG with a wide range of clinical applications from Alzheimer disease [61], chronic stroke [62] to human focal epilepsy [13]. Taking the whole-brain modeling approach used in TVB, the mean-field models of local neuronal dynamics can be combined with the structural information obtained from non-invasive brain imaging to simulate the spatiotemporal evolution of brain activity. This personalized strategy to construct a patient-specific whole-brain model allows testing of clinical hypotheses and exploration of novel therapeutic techniques to improve epilepsy surgery outcomes [13, 63, 64]. In our recent study [4], we have proposed a probabilistic framework, namely the Bayesian Virtual Epileptic Patient (BVEP), which provides an appropriate form of reparameterization to estimate the spatial map of epileptogenicity for an individualized whole-brain model of epilepsy spread. The BVEP model is equipped with a generative model capable of realistically producing various patterns of pathological brain activity during epileptic seizures [28, 65]. However, different EZ hypotheses can be confronted directly against imaging data in the clinical practice [13]. Using the BVEP model, the EZ hypothesis evaluation can be improved by incorporating different hypotheses as prior knowledge in the generative model. This approach then can be further tested via fully Bayesian criteria in a simulation paradigm as we demonstrate in this study. Systematic detection of EZ in the BVEP brain model aids in providing a patient-specific strategy for clinical hypothesis testing about EZ location to improve surgery outcomes.

In this study, we aim to provide predictive insight into the performance of structural and functional brain network organizations in explaining the individual's spatiotemporal brain activity at the macroscopic level. To this end, a patient-specific whole-brain model of epilepsy spread is inverted to estimate the spatial distribution of epileptogenicity across different brain areas. Based upon the single point estimates and Markov Chain Monte Carlo (MCMC [15]) sampling in the spirit of Bayesian inference, we then assess the model's predictive accuracy to

evaluate the main components of the large-scale brain network model that vary across patients (e.g., the patient-specific connectome as the structural component, and the spatial map of epileptogenicity as the functional component). In particular, we investigate how the prior information about model parameters affects the statistical power of fully Bayesian criteria as a selection rule to determine the most likely among a set of potential hypotheses.

Using in-silico data generated by TVB, and High Performance Computing (HPC) managed by JURECA booster in the Jülich Supercomputing Centre [66], this study demonstrates the limitation of classical information criteria in EZ hypothesis testing. We show that by taking into account the prior anatomical knowledge, the fully Bayesian criteria such as WAIC and LOO cross-validation provide decisive evidence regarding the degree of epileptogenicity across different brain regions. Our results illustrate that significant changes in WAIC and LOO cross-validation can be observed when the prior knowledge on the spatial epileptogenicity of brain regions is informative, whereas there is still support in favor of competing hypotheses if the prior is uninformative. Moreover, by placing an informative or a weakly informative prior distribution on the global coupling parameter describing the scaling of the brain's structural connectivity (SC), both WAIC and LOO are able to distinguish subject-specific SC from a set of candidates. We also illustrate that the fully Bayesian information-based criteria perform equally well as cross-validation technique in our simulated conditions. Collectively, our results indicate that the fully Bayesian criteria such as WAIC and LOO cross-validation provide an appropriate selection rule for determining the most likely among a set of competing hypotheses regarding both structural (the patient's connectome) and functional (the spatial map of epileptogenicity) components of the large-scale brain network models that vary across subjects. The framework we present here leverages the power of HPC to efficiently perform the computationally intensive inference. This compatibility can also be used to deploy parallel brain tractography, simulations, and model validation for different patients to provide swift and reliable supporting information to clinicians.

## Materials and methods

The main body of the work is based on the BVEP, a platform-independent probabilistic brain network approach, which enables us to infer the spatial map of epileptogenicity in a personalized large-scale brain model of epilepsy spread [4]. As schematically illustrated in Fig 1, our workflow to build the generative BVEP model for the purpose of epileptogenicity hypothesis testing comprises the following steps: at the first step, the non-invasive brain imaging data such as MRI and diffusion-weighted MRI (dwMRI) are collected for a specific patient. Using the TVB-specific reconstruction pipeline [28, 63, 67], the brain is parcellated into different regions which constitute the brain network nodes. The SC matrix, whose entries represent the connection strength between the brain regions, is derived from diffusion MRI tractography. This step constitutes the structural brain network model, which imposes a constraint upon network dynamics for the given subject. Then, a neural mass model of local brain activity is placed at each brain region. Each region is connected to other regions via the patient-specific SC matrix. This combination of brain's anatomical information with the mathematical modeling of population-level neural activity constitutes the functional brain network model, which allows the simulation of various spatiotemporal patterns of brain activity as observed in some brain disorders [13, 63]. In the BVEP, Epileptor model [65] is placed at each brain network node to realistically reproduce the onset, progression and offset of seizure patterns across different brain regions [4]. Subsequently, the functional brain network organization is integrated with the spatial map of epileptogenicity across different brain regions (i.e., different hypotheses on the location of EZ) to build the generative Virtual Epileptic Patient (VEP) brain model. The

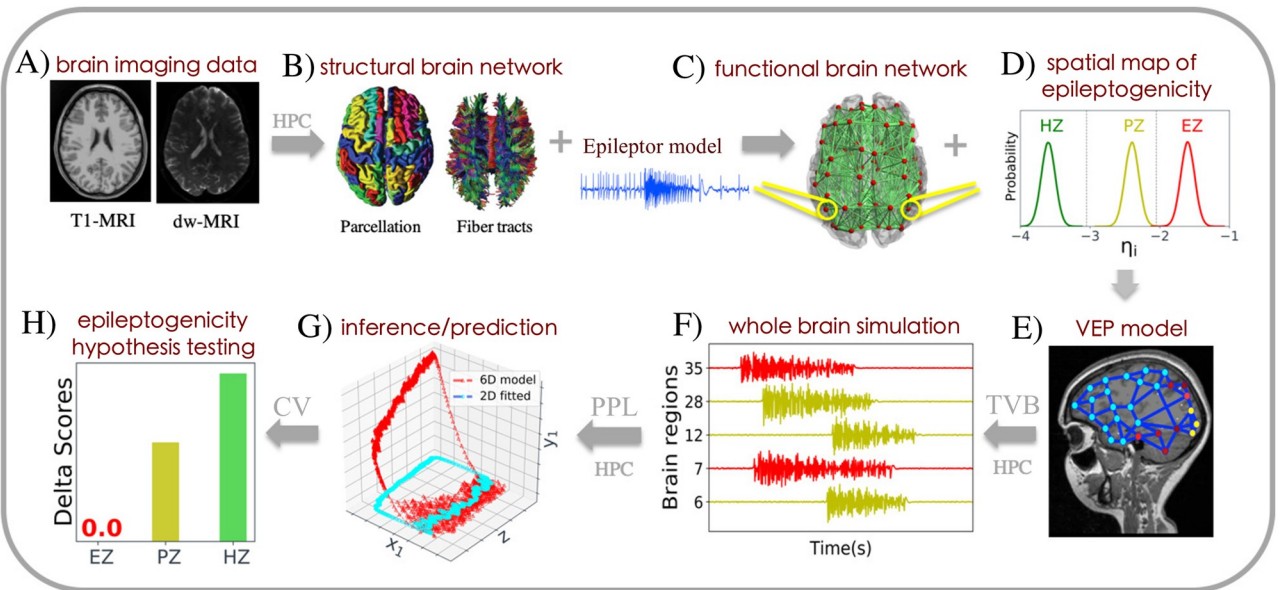

**Fig 1. The workflow of BVEP aims at assessing different hypotheses regarding the location of an epileptogenic zone in a personalized whole-brain model of epilepsy spread.** (**A**) At the first step, the patient undergoes non-invasive brain imaging such as MRI, and diffusion-weighted MRI (dwMRI). (**B**) Based on these images, the structural brain network model including brain parcellation and fiber tracts of the human brain are generated. (**C**) Then, a neural mass model (here 6D Epileptor) is defined at each brain region and connected through the structural connectivity matrix to build the functional brain network model. (**D**, **E**) Next, the VEP brain model is constructed by furnishing the functional brain network model with the spatial map of epileptogenicity (EZ, PZ, HZ hypotheses) across different brain regions. (**F**) The simulations by TVB allow to specifically mimic the individual's spatiotemporal macro-level brain activity. (**G**) Subsequently, the generative VEP brain model is embedded within a PPL tool to invert the BVEP brain model (here 2D Epileptor) and evaluating the model against the patient's data. (**H**) Finally, cross-validation (CV) is performed using WAIC/LOO to assess the model's prediction performance on new data, in order to evaluate and validate the quality of the hypotheses regarding epileptogenicity.

later simulations allow to mimic outputs of experimental modalities such as electro- and magnetoencephalography (EEG, MEG) and functional Magnetic Resonance Imaging (fMRI). Following the virtual brain reconstruction, the BVEP is constructed by embedding the generative VEP brain model within a probabilistic programming language (PPL) tool such as Stan [68] or PyMC3 [39] to fit and validate the simulations against the patient-specific data. Stan is a state-of-the-art platform for statistical modeling and high-performance statistical computation which can be run in popular computing environments (see https://mc-stan.org). PyMC3 provides several MCMC algorithms for model specification directly in Python code (see https://docs.pymc.io). In the present work, for inversion of BVEP model, we used the No-U-Turn Sampler (NUTS [19, 38]), a self-tuning variant of Hamiltonian Monte Carlo (HMC [69, 70]). Finally, cross-validation can be performed by WAIC or approximate LOO from the existing samples of log-likelihood to assess the model's ability in new data prediction. This approach enables us to efficiently evaluate different hypotheses about the location of EZ and further inform the expert clinician prior to epilepsy surgery.

## Individual patient data

For this study, we selected a patient (a 35 year-old male) initially diagnosed with left temporal lobe epilepsy (Histopathology: Gliosis, Surgical procedure: resection, Surgical outcome: seizure free, Engel score I). The patient underwent comprehensive presurgical evaluation, including clinical history, neurological examination, neuropsychological testing, structural and diffusion MRI scanning, Stereotactic-EEG (SEEG) recordings along with video monitoring as previously described in [28, 30]. The evaluation included non-invasive T1-weighted imaging (MPRAGE

sequence, repetition time = 1900 ms, echo time = 2.19 ms, 1.0 x 1.0 x 1.0 mm, 208 slices) and diffusion MRI images (DTI-MR sequence, angular gradient set of 64 directions, repetition time = 10.7 s, echo time = 95 ms, 2.0 x 2.0 x 2.0 mm, 70 slices, b-weighting of 1000 smm$^{-2}$). The images were acquired on a Siemens Magnetom Verio 3T MR-scanner.

## Stereotactic-EEG (SEEG) data preprocessing

For the selected patient, nine SEEG electrodes were placed in critical regions based on the presurgical evaluation. SEEG electrodes comprise 10 to 15 contacts. Each contact is 2 mm of length, 0.8 mm in diameter, and is 1.5 mm apart from other contacts. Brain signals were recorded using a 128-channel Deltamed system (sampling rate: 512 Hz, hardware band-pass filtering: between 0.16 and 97 Hz). The SEEG data is re-referenced using a bipolar montage, which is obtained using the difference of 2 neighboring contacts on one electrode. We extract the bipolarized SEEG signal from 5s before seizure onset up to 5s after seizure offset. The onset and offset times of the epileptic seizure are set by clinical experts. In this study, the log power of high frequency activity is used as the target for the fitting task. More precisely, the SEEG data are windowed and Fourier transformed to obtain estimates of their spectral density over time. Then SEEG power above 10 Hz is summed to capture the temporal variation of the fast activity [13, 28]. The envelope is calculated using a sliding-window approach with a window length of 100 time points. The signal inside the window is squared, averaged and log transformed. From the resulting envelope, we identify and remove outliers. Finally, the envelope is smoothed using a lowpass filter with a cut-off in the range of 0.05 Hz. The mean across the first few seconds of the envelope is used to calculate a baseline which is then subtracted from the envelope. Contacts are selected according to the mean energy of bipolars to provide a bijection map between source activity at brain regions and measurement at electrodes.

## Structural brain network model

The structural connectome was built with the TVB-specific reconstruction pipeline using generally available neuroimaging software [28, 63, 67]. First, the command *recon-all* from the Freesurfer package [71] in version v6.0.0 was used to reconstruct and parcellate the brain anatomy from T1-weighted images. Then, the T1-weighted images were coregistered with the diffusion weighted images by the linear registration tool *flirt* [72] from the FSL package in version 6.0 using the correlation ratio cost function with 12 degrees of freedom.

The MRtrix package in version 0.3.15 was then used for the tractography. The fibre orientation distributions were estimated from DWI using spherical deconvolution [73] by the *dwi2fod* tool with the response function estimated by the *dwi2response* tool using the *tournier* algorithm [74]. Next, we used the *tckgen* tool, employing the probabilistic tractography algorithm iFOD2 [75] to generate 15 millions fiber tracts. Finally, the connectome matrix was built by the *tck2connectome* tool using the Desikan-Killiany parcellation [76] generated by FreeSurfer in the previous step. The connectome was normalized so that the maximum value is equal to one.

## Functional brain network model

To build the structural brain network model, the brain is parcellated into different regions, which constitute the brain network nodes. In addition, the network connectivity is derived from diffusion weighted imaging that estimates the density of white matter tracts between brain regions. Then, a set of nonlinear stochastic differential equation is used to model the population-level brain activity at each node of the parcellation. The combination of structural data of individuals with the dynamical properties constitutes the functional brain network

model allowing to mimic the whole-brain activity for a specific subject as observed in neuro-imaging recordings such as EEG, MEG and fMRI. In the VEP model, a neural mass model of brain pathological patterns known as Epileptor [65] is placed at each brain network node to realistically reproduce the dynamics of onset, progression and offset of seizure-like events [65, 77]. Following Jirsa *et al.* [13], N-Epileptors are coupled through SC matrix to build the full VEP brain model:

$$
\begin{aligned}
\dot{x}_{1,i} &= y_{1,i} - f_1(x_{1,i}, x_{2,i}) - z_i + I_1 \\
\dot{y}_{1,i} &= \frac{1}{\tau_1}(1 - 5x_{1,i}^2 - y_{1,i}) \\
\dot{z}_i &= \frac{1}{\tau_0}(4(x_{1,i} - \eta_i) - z_i - K\sum_{j=1}^{N} C_{ij}(x_{1,j} - x_{1,i})) \\
\dot{x}_{2,i} &= -y_{2,i} + x_{2,i} - x_{2,i}^3 + I_2 + 0.002g(x_{1,i}) - 0.3(z_i - 3.5) \\
\dot{y}_{2,i} &= \frac{1}{\tau_2}(-y_{2,i} + f_2(x_{2,i}))
\end{aligned}
\tag{1}
$$

where

$$
f_1(x_1, x_2) = \begin{cases} x_1^3 - 3x_1^2 & \text{if } x_1 < 0 \\ (x_2 - 0.6(z - 4)^2)x_1 & \text{if } x_1 \geq 0 \end{cases}
$$

$$
f_2(x_2) = \begin{cases} 0 & \text{if } x_2 < -0.25 \\ 6(x_2 + 0.25) & \text{if } x_2 \geq -0.25 \end{cases}
$$

$$
g(x_1) = \int_{-t_0}^{t} \exp^{-\gamma(t-\tau)} x_1(\tau) dt,
$$

with $\tau_0 = 2857$, $\tau_1 = 1$, $\tau_2 = 10$, $I_1 = 3.1$, $I_2 = 0.45$, and $\gamma = 0.01$. Here, $i \in \{1, 2, \ldots, N\}$, where $N$ is the number of brain regions. The coupling between brain areas is defined by a linear approximation $K\sum_{j=1}^{N} C_{ij}(x_{1,j} - x_{1,i})$, which includes a global scaling factor $K$ and the patient's connectome $C_{ij}$. In the VEP brain model, the variables $x_1$ and $y_1$ account for the fast discharges during the ictal seizure state, whereas the variables $x_2$ and $y_2$ represent the slow spike and wave events observed in electrographic seizure recordings [65]. These variables are linked together by the permittivity state variable $z$, which is responsible for the transition between interictal and ictal states. Moreover, the degree of epileptogenicity across brain regions is represented via the value of excitability parameter $\eta$. If $\eta > \eta_c$, where $\eta_c$ is the critical value of epileptogenicity, the brain region shows seizure activity autonomously and is referred to as epileptogenic; otherwise it is in its (healthy) equilibrium state and does not trigger seizures autonomously.

Under the assumption of time scale separation and focusing on the slower time scale, the effect of variables $x_2$ and $y_2$ are negligible by averaging [77], while the fast variables ($x_1$ and $y_1$) rapidly collapse on the slow manifold [78], whose dynamics is governed by the slow variable $z$. This approach yields the 2D reduction of VEP model as follows:

$$
\begin{aligned}
\dot{x}_{1,i} &= 1 - x_{1,i}^3 - 2x_{1,i}^2 - z_i + I_{1,i} \\
\dot{z}_i &= \frac{1}{\tau}(4(x_{1,i} - \eta_i) - z_i - K\sum_{j=1}^{N} C_{ij}(x_{1,j} - x_{1,i})).
\end{aligned}
\tag{2}
$$

Finally, the BVEP model is constructed by embedding the generative VEP brain model within a PPL tool such as Stan/PyMC3 to fit and validate the model against the patient's data. In this study, we have simulated the full VEP brain model by The Virtual Brain (TVB [59]) platform to generate the synthetic data, whereas the 2D reduction of VEP model was used to fit the envelope of time series and infer the excitability parameters $\eta_i$ [4].

Note that, the full VEP model comprises five state variables governed by three different timescales (Eq 1) and is a complete taxonomy of epileptic seizures, which demonstrate a thorough description of system bifurcations that give rise to onset, offset and seizure evolution [65, 79]. Whereas, the 2D reduced variant of the VEP (Eq 2) is a fast-slow dynamical system to modeling the average of fast discharges during the ictal seizure states and the seizure propagation. This reduction is limited to demonstrate only saddle-node bifurcation out of six types of bifurcations involved in the spatiotemporal seizure dynamics. However, the seizure initiation from EZ and propagation to PZ are invariant under this model reduction [28, 77]. The 2D VEP model can successfully predict the key data features such as onset, propagation envelope and offset, while considerably alleviating the computational burden of the Bayesian inversion [4, 13].

## Epileptogenicity hypothesis formulation

In addition to the brain's anatomical information which structurally impose a constraint upon network dynamics [80–82], the individual variability can be further constrained by EZ hypothesis formulation in order to produce more specific pathological patterns of brain activity. The hypothesis on the location of EZ allows refining the network pathology to better predict seizure onset and propagation for a given patient.

In the BVEP generative model, the capacity of each brain region to trigger seizures depends on its connectivity and the excitability parameter, which is then the target of parameter inference. Depending on the excitability value $\eta$, different brain regions are classified into three main types:

- Epileptogenic Zone (EZ): if $\eta > \eta_c$, where the brain region can trigger seizures autonomously (responsible for the origin and early organization of the epileptic activity). Within the Bayesian framework, an EZ hypothesis can be defined by a normal distribution $\mathcal{N}(\eta_{ez}, \sigma^2)$ with $\eta_{ez} > \eta_c$, where $\sigma$ represents the degree of uncertainty about this hypothesis that the brain region is epileptogenic.

- Propagation Zone (PZ): if $\eta_c - \Delta\eta < \eta < \eta_c$, then the brain region does not trigger seizures autonomously but it may be recruited during the seizure evolution, since the corresponding equilibrium state is close to the critical value. In the Bayesian environment, a PZ hypothesis can be formulated by $\mathcal{N}(\eta_{pz}, \sigma^2)$, with $\eta_c - \Delta\eta < \eta_{pz} < \eta_c$, where $\sigma$ determines the level of our information about this hypothesis that the brain region is recruited during the seizure evolution.

- Healthy Zone (HZ): if $\eta < \eta_c - \Delta\eta$, where the brain region stays away from triggering seizures autonomously. In the Bayesian approach, an HZ hypothesis can be codified by $\mathcal{N}(\eta_{hz}, \sigma^2)$, with $\eta_{hz} < \eta_c - \Delta\eta$, where $\sigma$ indicates the amount of uncertainty about this hypothesis that the brain region is healthy.

Here, we assume that seizures originate from a subnetwork(s) denoted by EZ, and recruit a secondary subnetwork(s) denoted by PZ that are strongly coupled to the EZ. In other words, seizure initiation is a region's intrinsic property and depends only on the excitability parameter of the brain regions, whereas seizure propagation is a complex network effect which

depends on the interplay between multiple factors including the brain region's excitability (node dynamics) [30, 83, 84], the individual structural connectivity (network structure) [28, 32, 34], and brain state dependence (network dynamics).

## Bayesian inference

The Bayesian approach offers a framework to deal with parameter estimation and model uncertainty. The parameter estimation within a Bayesian framework is treated as the quantification and propagation of uncertainty, defined via a probability, in the light of observations [4]. The uncertainty over the range of possible parameter values is also estimated, rather than a single point estimate (e.g., the maximum likelihood estimate) in the frequentist approach [3].

The Bayesian inference is based on prior knowledge and the likelihood function of model parameters [15]. The prior distribution $p(\theta)$ includes our best knowledge about possible values of the parameters, whereas the likelihood $p(y \mid \theta)$ represents the probability of obtaining the observation $y$, with a certain set of parameter values $\theta$. Through the Bayes' rule, the prior and the likelihood combinedly define a posterior distribution $p(\theta \mid y)$, which represents the actual parameter distribution conditioned on the observed data:

$$p(\theta \mid y) = \frac{p(y \mid \theta)p(\theta)}{p(y)}, \tag{3}$$

where $p(y) = \int p(y \mid \theta)p(\theta)d\theta$ denotes model evidence (or marginal likelihood), which in the context of inference amounts to simply a normalization term.

Considering the 2D reduction of the VEP model (cf., Eq 2), then $\theta = (x_{1,1}(t_0), x_{1,2}(t_0), \ldots, x_{1,N}(t_0), z_1(t_0), z_2(t_0), \ldots, z_N(t_0), \eta_1, \eta_2, \ldots, \eta_N, K, \sigma, \sigma') \in \mathbb{R}^k$, where $k = 3N + 3$, and $N$ is equal to the number of brain regions (here $N = 84$). For $i \in \{1, 2, \ldots, N\}$, the parameters $x_{1,i}(t_0)$ and $z_i(t_0)$ indicate the initial state of the 2D reduced VEP model, and $\eta_i$ indicates the excitability parameter of each brain region. Moreover, the hyper-parameters $\sigma$ and $\sigma'$ represent the standard deviation of the process (dynamical) noise and the measurement/observation noise, respectively.

In order to compute the posterior distributions, we employ Markov Chain Monte Carlo (MCMC [15]) methods to construct a Markov chain whose steady-state distribution asymptotically approaches the distribution of interest. The MCMC approach has the advantage of being nonparametric and asymptotically exact in the limit of long/infinite runs [14].

It is widely known that gradient-free MCMC algorithms such as Metropolis-Hastings, Gibbs sampling, and slice-sampling often do not sample efficiently from posterior distributions in complex and nonlinear inverse problems, specifically when the model's parameters are highly correlated [19, 85, 86]. In contrast, gradient-based MCMC algorithms such as Hamiltonian Monte Carlo (HMC [70]) use the gradient information of the posterior to avoid the naive random walk of the traditional sampling algorithms, thereby sample efficiently from posterior distributions with correlated parameters [4, 19]. However, HMC requires a careful tuning of the algorithm parameters to ensure efficient sampling from posterior distributions [21, 87]. The recently developed MCMC algorithm known as No-U-Turn Sampler (NUTS [19]), a self-tuning variant of HMC [38], solves these issues by adaptively tuning the parameters of the algorithm to efficiently sample from complex posterior distributions. Using a recursive algorithm, NUTS eliminates the need to set the number of leapfrog steps that the algorithm takes to generate a proposal state. Taking the advantage of automatic differentiation, it samples efficiently from analytically intractable posterior distributions with a high degree of curvature [18, 88].

## Information criteria

The predictive accuracy is one of the criteria that can be used to evaluate the overall fit of a model to the observed data and thus as a selection rule to compare different candidate models, even if all of the models represent mismatches with the data [45]. To assess the predictive accuracy, the log-likelihood, $\log p(y \mid \theta)$, has been widely used in the statistical literature as the basis of information criteria (the accuracy term) with subtraction of an approximate bias correction (the penalty term) [48]. Classical information criteria are model quality measures that only depend on the maximum log-likelihood (MLL), as a measure of model accuracy. In addition, the number of data points and/or the number of model parameters quantify the measure of model complexity [45].

A commonly used model comparison measure is Bayesian information criterion (BIC [51])

$$\text{BIC} = -2 \log p(y \mid \hat{\theta}_{\text{mle}}) + k \log(n), \tag{4}$$

where $\hat{\theta}_{mle}$ is the maximum likelihood estimate (MLE), and the term $\log p(y \mid \hat{\theta}_{mle})$ indicates the MLL based on $\hat{\theta}_{mle}$. Here, $k$ is the number of free model parameters, and $n$ is the number of data points. Among a set of candidate models, the best model is the one that provides the minimum of BIC.

Another popular information-criterion-based model comparison method is Akaike's information criterion (AIC [50])

$$\text{AIC} = -2 \log p(y \mid \hat{\theta}_{\text{mle}}) + 2k. \tag{5}$$

Empirically, BIC is observed to be biased towards the simple models and AIC to the complex models [23, 24].

The corrected AIC [89, 90], an improved version of AIC for finite-sample calculations is given by

$$\text{AICc} = \text{AIC} + \frac{2k(k+1)}{n-k-1}. \tag{6}$$

The corrected AIC improves the AIC for small samples, while it approximates the AIC for large samples with $n > k^2$. For $n < k + 1$, the denominator in the correction term becomes negative and AICc penalizes parameters less than AIC does [23, 46]. In this work, we do not use AICc, since $n > k^2$, where AICc closely approximates the AIC.

In contrast to the classical information criteria such as BIC and AIC, which involved calculation of the point estimates, Watanabe-Akaike (or widely available) information criterion (WAIC [58]) and leave-one-out (LOO [57]) cross-validation use the whole posterior distribution and are considered as fully Bayesian methods for estimating the pointwise out-of-sample prediction accuracy [55, 56]. In practice, the computation of WAIC and LOO may involve additional computational steps, however, using the new approaches that benefit from the existing posterior samples [55], their computation time is negligible compared to the cost of model fitting. WAIC and an importance-sampling approximated LOO can be estimated directly using the log-likelihood evaluated at the draws from posterior distributions of parameters. [55, 68]. Because WAIC and LOO are fully Bayesian criteria, they have many advantages over the classical information criteria [45]. For instance, they enable us to take into account the prior knowledge about the model parameters and thus the choice of the prior can improve the resulting model comparisons. Moreover, unlike other information criteria, WAIC is invariant to parameterization and is also defined for singular models (i.e. models with Fisher information matrices that may fail to be invertible). In fact, WAIC is based on the series expansion of

LOO, and asymptotically they are equal [55, 56]. However, it has been reported that LOO is more robust in the finite case with weak priors or influential observation [55].

The aim of using WAIC and LOO is to estimate the accuracy of predictive distribution $p(\tilde{y}_i \mid y) = \int p(\tilde{y}_i \mid \theta)p(\theta \mid y)d\theta$, where $y$ is the observed data and $\tilde{y}_i$ indicates the future (unobserved) data. Following Gelman *et al.* [45], WAIC is given by

$$\text{WAIC} = -2(\text{lppd}_{\text{waic}} - \text{p}_{\text{eff}}), \qquad (7)$$

where $lppd_{waic}$ is the log pointwise predictive density, and $p_{eff}$ is the estimated effective number of parameters. The $lppd_{waic}$ of observed data $y$ is an overestimate of the mean log predictive density for a new data set $\tilde{y}_i$. To measure the predictive accuracy of the fitted model, $lppd_{waic}$ is defined as [45]

$$
\begin{aligned}
lppd_{waic} &= \sum_{i=1}^{n} \log p(y_i \mid y) \\
&= \sum_{i=1}^{n} \log \int p(y_i \mid \theta)p_{post}(\theta)d\theta,
\end{aligned}
\qquad (8)
$$

where $p_{post}(\theta) = p(\theta \mid y)$ is the posterior distribution of parameters. Here, $\log p(y_i \mid y) = \log\int p(y_i \mid \theta)p(\theta \mid y)d\theta$, assuming that the data $y$ is divided into $n$ individual points $y_i$. The $lppd_{waic}$ is fully Bayesian because of $p_{post}(\theta)$, which can be viewed as a Bayesian analogue of the term $\log p(y \mid \hat{\theta}_{mle})$ used in the computation of AIC and BIC.

In practice, we can replace the expectations by the average over the draws $\theta^s$, $s = 1, \ldots, S$ from the full posterior $p(\theta \mid y)$. According to Gelman *et al.* [45], $lppd_{waic}$ can be computed from the posterior samples with the following equation

$$lppd_{waic} = \sum_{i=1}^{n} \log \left(\frac{1}{S}\sum_{i=1}^{S} p(y_i \mid \theta^s)\right). \qquad (9)$$

Moreover, by summing over all the posterior variance of the log predictive density for each data point $y_i$, the estimated effective number of parameters as a measure of model complexity can be computed from

$$p_{eff} = \sum_{i=1}^{n} V_{s=1}^{S}\left(\log p(y_i \mid \theta^s)\right), \qquad (10)$$

where $V_{s=1}^{S}$ represents the sample variance i.e., $V_{s=1}^{S}(a_s) = \frac{1}{S-1}\sum_{s=1}^{S}\left(a_s - \bar{a}\right)^2$. Here, $\log p(y \mid \theta)$ indicates the log predictive density, which in practice has the dimension of $S \times n$, where $S$ is the number of samples from posterior distribution, and $n$ is the number of data points.

Note that the computation of both $lppd_{waic}$ and the penalty term $p_{eff}$ in WAIC uses the whole posterior distribution rather than point estimates $\theta_{mle}$ in the computation of AIC and BIC. Similar to BIC and AIC, lower values of WAIC indicate a better model fit.

Another fully Bayesian method to estimate predictive accuracy for unobserved data is leave-one-out (LOO [57]) cross-validation. Computing LOO requires taking out one data point at a time and refitting the model with the remaining training sets. Bayesian LOO estimate of out-of-sample predictive fit denoted by $lppd_{loo}$ can be computed based on the

definition [45]

$$
\begin{aligned}
lppd_{loo} &= \sum_{i=1}^{n} \log p(y_i \mid y_{-i}) \\
&= \sum_{i=1}^{n} \log \int p(y_i \mid \theta) p_{post(-i)}(\theta) d\theta,
\end{aligned}
\tag{11}
$$

where $p_{post(-i)}$ is the posterior density given the data without the $i$th data point.

Due to the iterative nature of cross-validation, the exact computation of LOO can be prohibitive for large sample data sets (it requires to fit the model $n$ times, where $n$ is the number of trials). A relatively new importance sampling technique, Pareto Smoothed Importance Sampling (PSIS [56]), has been proposed for stabilizing importance weights, which enables us to efficiently approximate LOO cross-validation without refitting the model with different training sets. Using the posterior draws $\theta^s$, the PSIS estimate of $lppd_{loo}$ can be computed with the following equation [55]

$$
lppd_{loo} = \sum_{i=1}^{n} \log \left( \frac{\sum_{i=1}^{S} w_i^s p(y_i \mid \theta^s)}{\sum_{i=1}^{S} w_i^s} \right),
\tag{12}
$$

where, $w_i^s, s = 1, \ldots, S$ is a vector of importance weights for $i$th data point. Finally, Bayesian LOO cross-validation requires no penalty term and it is defined as $-2lppd_{loo}$ to be on the same scale as WAIC. In this study, we use PSIS to estimate Bayesian leave-one-out cross-validation.

## Delta score

Typically, the relative differences in information criteria is used to measure the level or the strength of evidence for each candidate model. In this study, for each of BIC/AIC/WAIC/LOO, the Delta score is calculated to determine the level of support for each fitted model. Delta score is the relative difference in information criteria between the best model (which has a Delta score of zero) and each other candidate model in the set of consideration. For each of BIC/AIC/WAIC/LOO, the Delta score of j-th candidate model denoted by $\Delta IC_j$ is defined by:

$$
\Delta IC_j = IC_j - IC_{min},
\tag{13}
$$

where $IC_j$ indicates the value of BIC/AIC/WAIC/LOO for j-th candidate model, and $IC_{min}$ is the minimum of $IC_j$ values (i.e., the value of information criterion for the best model). A larger value of $\Delta IC_j$ indicates that j-th model is less plausible. Following Burnham and Anderson [48], a $\Delta IC_j \leq 2$ suggests substantial evidence to support j-th model, if $2 \leq \Delta IC_j \leq 4$ then there is strong support for j-th model. If $4 \leq \Delta IC_j \leq 7$ then the model has considerably less support, whereas a $\Delta IC_j > 10$ indicates that the model is very unlikely (i.e., the model has essentially no support). In the case of epileptogenicity hypothesis testing, $j \in \{EZ, PZ, HZ\}$, the best hypothesis between EZ, PZ, and HZ has a Delta score of zero.

## Inference diagnostics

Once the model parameters have been estimated, it is necessary to assess the convergence of MCMC samples. A quantitative way to assess the MCMC convergence to the stationary distribution is to calculate the potential scale reduction factor $\hat{R}$ [91, 92], which provides an estimate of how much variance could be reduced by running chains longer. Each model parameter/hidden state has $\hat{R}$ statistic associated with it, which is based on the variance of the parameter as a

weighted sum of the within-chain and between-chain variance [91, 92]. When $\hat{R}$ is approximately less than 1.1, the MCMC convergence has been achieved (approaches to 1 in the case of infinite samples); otherwise, the chains need to be run longer to improve convergence to the stationary distribution [14]. In this study, for each HMC chain, the $\hat{R}$ diagnostic is monitored to check whether the convergence has been achieved. Moreover, the NUTS-specific diagnostics such as the average acceptance probability of samples, the number of divergent leapfrog transitions, the step size used by NUTS in its Hamiltonian simulation, and the tree depth used by NUTS have been monitored to validate the estimates.

## Evaluation of posterior fit

In order to quantify the accuracy of the estimated spatial map of epileptogenicity across different brain regions, we plot the posterior z-scores (denoted by $\zeta_i$) against the posterior shrinkages (denoted by $\rho_i$) defined by:

$$\zeta_i = |\frac{\bar{\eta}_i - \eta_i^*}{\sigma_{post}}|, \tag{14}$$

$$\rho_i = 1 - \frac{\sigma_{i,post}^2}{\sigma_{i,prior}^2}, \tag{15}$$

where $\bar{\eta}_i$ and $\eta_i^*$ are the estimated-mean and the ground-truth of each brain node's excitability parameter, respectively, whereas $\sigma_{i,prior}^2$, and $\sigma_{i,post}^2$ indicate the variance of prior and posterior distribution related to each excitability estimation, respectively. In this study, $i \in \{1, 2, \ldots, N\}$, with $N = 84$ indicating the number of brain regions included in the analysis. The posterior z-score quantifies how much the posterior distribution encompasses the ground truth, while the posterior shrinkage quantifies how much the posterior distribution contracts from the initial prior distribution [93].

## High performance computing for Bayesian inversion

Bayesian inversion is amenable to "embarrassingly parallel" computational approaches which benefit greatly from multi-core supercomputer parallelization. Here embarrassingly parallel means that the computation may be decomposed into completely independent sub-computations that can then be cheaply recomposed as final aggregate values (such as mean, variance, and so on); this implies that for the bulk of the computation, the only limitations to speeding up the wall-time to solution is the available raw number of parallel units (CPU cores, GPU cores) and the minimum quantum of the computation. Recently, several approaches to Bayesian inversion have leveraged the parallelism offered by new computational architectures such as multi-core CPUs and GPUs to enhance the wall-time to solution of Monte Carlo simulations [94–96]. Previously, Angelikopoulos *et al.* [95] proposed a framework to parallelize the generation, sampling, and evaluation of Markov chains on HPC using a master-worker approach. More recently, Mahani *et al.* [94] proposed two types of optimizations that leverage parallelized MCMC using SIMD (Single Instruction Multiple Data) approaches amenable to implementation on multi-core processors and in GPUs. The first optimization targeted concurrent sampling of conditionally-independent nodes; the second consisted of calculating the contributions from child-nodes to the posterior of each node concurrently. Although Mahani *et al.* [94] focused particularly on performance for Bayesian networks, both optimizations are applicable more generally for parallelizing MCMC on multi-core processors and GPUs.

The parallel execution of single chains can benefit from SIMD vectorization techniques. SIMD techniques optimize the execution of identical operations applied to different data, leveraging vectorizing hardware architectures such as GPUs and the Intel Xeon Phi 7250-F Knights Landing (KNL) CPUs [97]. Each KNL CPU has 68 cores with two Vector Processing Units (VPU), specialized electronic circuitry which can perform single operations simultaneously on all data points stored in a vector register, and a 512-bit vector register per core. Both VPUs can perform a multiply and add operation on the vector register per cycle, providing an ideal speedup proportional to the vector width [98]. The PPL tool used in this work to perform the Bayesian inference of the model parameters uses such vectorization to enhance the performance of the gradient calculations on the log-posterior [94, 99].

Furthermore, the execution of several computational Markov chains using independent random seeding is critical for statistically confirming the convergence of the model; these independent random chains can be computed on separate cores or nodes without computational interdependencies. The high degree of parallelization attainable on current HPC architectures increases the capability to detect problems with the convergence of the chains while keeping total execution time almost constant [100].

S1 Fig shows the scaling behaviour of executing an increasing number of Markov chains on an increasing number of computing nodes (1–32) on the KNL Booster partition of the JURECA supercomputer [66]. This test is also known as weak scaling and provides information about the maximum speedup which can be achieved for a workload of an increasing size on increasing number of computing units. It is possible to see that the total simulation time remains constant even when the number of chains increases (see S1(A) Fig), whereas the speedup achieved increases linearly with the number of computing nodes (see S1(B) Fig). This technique can be used not only to execute more chains for a single subject but it can also be used to perform inference on different subject data in parallel. By efficiently using HPC infrastructure, the work we present here allows faster Bayesian parameter estimation for individual models and enables its application for clinical and time-constrained purposes.

## Synthetic data and model inversion

The Virtual Brain (TVB [59]) is an open-access neuroinformatics platform written in Python to stimulate large-scale brain network models based on individual subject data. In this study, the VEP model simulation was performed within TVB using a Heun integration scheme (with the time step of 0.04 ms). A zero-mean white Gaussian noise with a variance of 0.0025 was added to the system state variables. Moreover, the initial conditions were selected randomly in the interval (−2.0, 5.0) for each state variable.

To simulate the seizure activity for a virtual patient, we selected two brain regions as EZ and three regions as PZ, defined at the nodes $EZ_{idx} \in \{7, 35\}$, and $PZ_{idx} \in \{6, 12, 28\}$, respectively. The excitability value was chosen as $\eta_{ez} = -1.6$ corresponding to regions in EZ, and $\eta_{pz} = -2.4$ for the regions in PZ, whereas all the other regions were defined as HZ with $\eta_{hz} = -3.6$.

Finally, Bayesian inference is performed by Stan using its command line interface [99] to invert the BVEP model for the simulated data sets. Simulations were run on the JURECA booster in the Jülich Supercomputing Centre [66]. The JURECA booster has a peak computing capacity of 5 petaflop/s, with 1,640 compute nodes using Intel Xeon Phi processors with 68 cores each. This multicore architecture allows the simulation of multiple parallel HMC chains, each on a separate processor. Typical simulations were deployed on 2040 cores and took in average 10 hours to execute. The code for simulations and posterior-based analysis was implemented in Python. The main source code repository is available at https://github.com/ins-amu/BVEP.

## Results

### The workflow of the BVEP for hypothesis testing

As schematically illustrated in Fig 1, our workflow to build the generative BVEP model for the purpose of epileptogenicity hypothesis testing comprises the following steps: at the first step, the non-invasive brain imaging data such as MRI and Diffusion-weighted MRI (dMRI) are collected for a specific patient (Fig 1A). Using HPC for parallel computation and TVB-specific reconstruction pipeline [28, 63, 67], the brain of each patient is parcellated into different regions which constitute the brain network nodes. The SC matrix, whose entries represent the connection strength between the brain regions, is derived from diffusion MRI tractography. This step constitutes the structural brain network model, which imposes a constraint upon network dynamics for the given subject (Fig 1B). Then, the Epileptor model [65] is placed at each brain network node to realistically reproduce the onset, progression, and offset of seizure patterns across different brain regions [65]. This combination of the brain's anatomical information with the mathematical modeling of population-level neural activity constitutes the functional brain network model, which allows the simulation of various spatiotemporal patterns of brain activity as observed in some brain disorders (Fig 1C). Subsequently, the functional brain network organization is integrated with the spatial map of epileptogenicity across different brain regions (i.e., different hypotheses on the location of EZ) to build the generative Virtual Epileptic Patient (VEP) brain model (Fig 1D and 1E). The forward models in TVB platform allow to mimic outputs of experimental modalities such as electro- and magnetoencephalography (EEG, MEG) and functional Magnetic Resonance Imaging (fMRI)(Fig 1F). Following the virtual brain simulations, the BVEP model is constructed by embedding the generative VEP brain model within a probabilistic programming language (PPL) tool such as Stan [68] or PyMC3 [39] to fit and validate the simulations against the patient-specific data (Fig 1G). In the present work, for inversion of the BVEP model, we used recently developed No-U-Turn Sampler (NUTS [19, 38]) algorithm, a self-tuning variant of Hamiltonian Monte Carlo (HMC [69, 70]), while High Performance Computing (HPC) allows us to run parallel MCMC for Bayesian inference. Finally, the cross-validation is performed by WAIC or approximate LOO from the existing samples of log-likelihood to assess the model's ability in new data prediction (Fig 1H). HPC on the JURECA booster in the Jülich Supercomputing Centre can operate in parallel the brain tractography, simulations, and model validation. This approach enables us to efficiently evaluate different hypotheses about the location of EZ and further inform the expert clinician prior to epilepsy surgery.

### Bayesian inference of the spatial map of epileptogenicity

First, we show the result of BVEP model inversion in estimating the spatial map of epileptogenicity across different brain areas. Fig 2A illustrates the constructed VEP brain model (see Eq 1) for a given patient. For this patient, two brain regions are defined as EZ (shown in red), three regions as PZ (in yellow), and all the other brain nodes are fixed as HZ (i.e., not epileptogenic, shown in green). Then, the VEP brain model is simulated by TVB, and we used Stan to infer the spatial distributions of excitability across different brain regions. Exemplary of the VEP model simulation for three brain node types EZ, PZ, and HZ versus the inferred envelope of time series is displayed in Fig 2B. The simulation illustrates the time series of fast activity variable in the full VEP model (i.e., $x_{1,i}(t)$ in Eq 1, shown by dash-dotted line), whereas the estimated envelope of time series is the result of inverting the reduced VEP model (cf. Eq 2, shown by dashed lines) using NUTS algorithm. From this figure, it can be seen that there is a remarkable similarity between the simulated and the predicted seizure activity regarding the

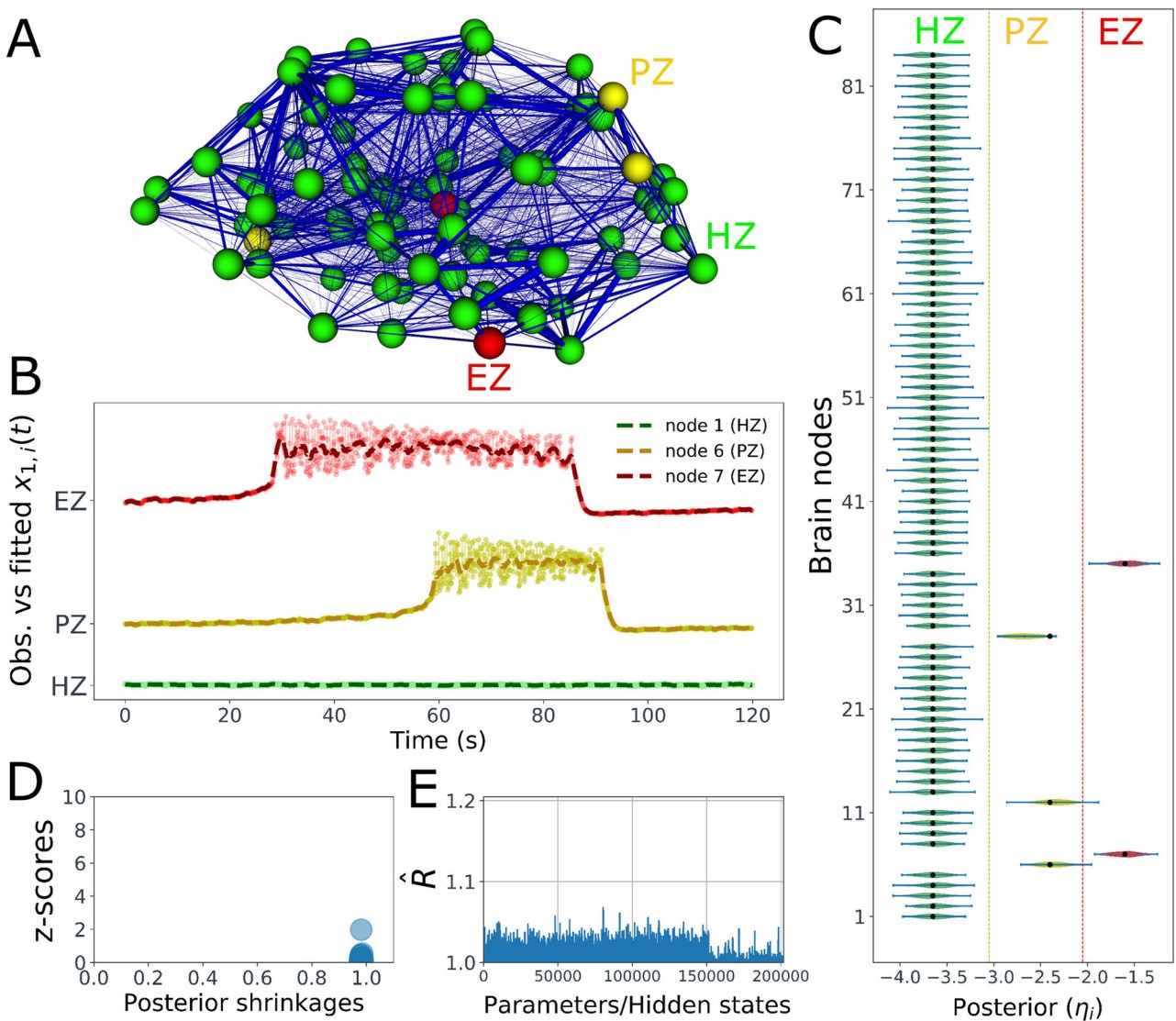

**Fig 2. Estimation of the spatial map of epileptogenicity across different brain regions in the BVEP model.** (**A**) The constructed VEP brain model; the patient-specific functional brain network model consisting of 68 cortical regions and 16 subcortical structures furnished with the spatial map of epileptogenicity (EZ, PZ, and HZ). Each brain region is represented as one node of the network, with color indicating its epileptogenicity (red: EZ, yellow: PZ, green: HZ). Thickness of the blue lines indicates the strength of the connections between brain regions (SC matrix). (**B**) Exemplary of the full VEP model simulation (dash-dotted lines) versus the fitted reduced VEP model (dashed lines) for three brain node types EZ, PZ, and HZ, shown in red, yellow, and green, respectively. (**C**) Violin plots of the estimated densities of excitability parameters for 84 brain regions included in the analysis. The ground truth values are displayed by the filled black circles. (**D**) The distribution of posterior z-scores and posterior shrinkages for all the inferred excitabilities confirms the accuracy and the reliability of the model inversion. (**E**) The values of $\hat{R}$ for all of the hidden states and parameters estimated by NUTS algorithm (lower than 1.05) implying that the HMC chain has converged.

initiation, propagation, and termination. Fig 2C shows the estimated posterior densities of the excitability parameter $\eta_i$ for all 84 brain regions included in the analysis. We observe that the true value of the excitability parameter (filled black circles) is well under the support of the posterior density for all the brain regions. See S2 Fig for two other estimations with different spatial maps of epileptogenicity across different brain areas. Also for a selected brain region, the model inversion by NUTS algorithm demonstrates an accurate and robust estimation by recovering the ground truth (see S4(A) Fig). Here, an identical weakly informative prior was

placed on all 84 brain regions included in the analysis i.e., $\mathcal{N}(\eta_i, \sigma^2)$, where $i \in \{1, 2, \ldots, 84\}$, with $\eta_i = -2.5$ and $\sigma = 1$ (see S3(**C**) Fig). Moreover, as shown in Fig 2**D**, the distribution of posterior z-scores and posterior shrinkages for all the inferred excitabilities confirms the accuracy and reliability of the Bayesian inversion. The concentration of posterior shrinkages towards one demonstrates that all the posteriors in the Bayesian inversion are well-identified, while the concentration of posterior z-scores towards zero implies that the true parameter values are accurately encompassed in the posteriors. Therefore, the distribution on the bottom right of the plot implies an ideal Bayesian inversion. Finally, in order to further confirm the reliability of the model inversion, the potential scale reduction factor $\hat{R}$, a reliable quantitative metric for MCMC convergence, is plotted in Fig 2**E**. It can be seen that the values of $\hat{R}$ for all of the hidden states and parameters estimated by NUTS algorithm are below 1.05 implying that the HMC chain has converged to the target distribution (the potential scale reduction factor $\hat{R}$ approaches to 1.0 in the case of infinite samples [14]). Here, by parcellating the brain into $N = 84$ regions and having 1200 data points per region as the observation, the Bayesian inversion involves estimating 201600 hidden states and 255 free parameters ($k = 3N + 3$).

## The effect of prior on the excitability estimation

To illustrate how fully Bayesian information criteria are affected by the parameters of a Gaussian prior placed on the excitability parameter of 2D Epileptor model (cf. Eq 2), here, WAIC and LOO are computed as a function of the mean and the standard deviation of prior. Fig 3 depicts the computed Delta scores for WAIC and LOO using different values of the mean of Gaussian prior, when the standard deviation of the prior on excitability parameter varies from very small to large values. More precisely, the prior placed on excitability parameter $\eta$ is assumed as a Gaussian distribution $\mathcal{N}(\mu, \sigma^2)$, in which $\mu$ accounts for the excitability value ranges from -4.5 to 1.5, whereas $\sigma$ representing the uncertainty about the estimation varies from 0.01 to 1000. In the simulation, the ground truth of excitability $\eta$ was -1.5, as illustrated by the dashed white line. The figure plots the averages over 4 HMC chains. From this figure, we observe that a small value of standard deviation ($\sigma \leq 0.1$) is required to distinguish different

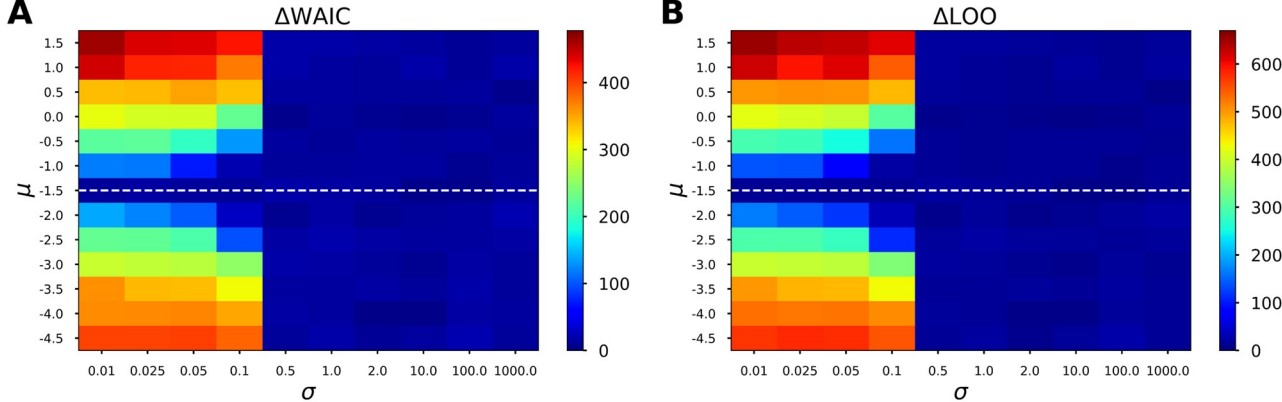

**Fig 3. Effect of the parameters of a Gaussian prior on the performance of WAIC and LOO in selecting the epileptogenicity hypothesis for the 2D Epileptor model.** To illustrate how fully Bayesian information criteria are affected by the parameters of a Gaussian prior placed on the excitability parameter, Delta scores for WAIC and LOO are computed as a function of the mean and the standard deviation of prior. Panels (**A**) and (**B**) show the computed $\Delta$WAIC and $\Delta$LOO, respectively, when the prior placed on excitability parameter is a Gaussian distribution $\mathcal{N}(\mu, \sigma^2)$, in which excitability value (denoted by $\mu$) varies from -4.5 to 1.5, whereas the uncertainty about the estimation (denoted by $\sigma$) ranges from 0.01 to 1000. It can be seen that WAIC closely approximates LOO cross-validation. The figure plots the averages over 4 HMC chains. The dashed white line indicates the true value of excitability parameter used in the simulation. A larger value of WAIC/LOO indicates that the parameter is less plausible.

excitability values used in the prior. In other words, if the prior placed on the excitability parameter is informative ($\sigma \leq 0.1$), then both WAIC and LOO are able to identify which value corresponds to the ground truth; otherwise, there is still evidence in favor of other values (in blue). Furthermore, comparing Fig 3A and 3B indicates that WAIC closely approximates the LOO cross-validation.

## Epileptogenicity hypothesis testing via information criteria

In order to illustrate the ability of information criteria as a selection rule for epileptogenicity hypothesis testing, three hypotheses as EZ, PZ, and HZ are evaluated on a specific brain region. Here, the brain area with the highest excitability value ($\eta_i$) corresponding to a region in EZ (node number 7) is selected, and for each of the EZ, PZ, and HZ hypothesis, the classical (BIC/ AIC), and the fully Bayesian (WAIC/LOO) information criteria are calculated based on the single point (MLE) and Bayesian estimates (MCMC), respectively.

In Fig 4, the computed BIC, AIC, WAIC, LOO, and their Delta scores (shown in cyan) averaged over 4 estimations are illustrated for each of the EZ, PZ, and HZ hypothesis. In addition, the deviance (in green), and the penalty term (in yellow) are displayed for BIC, AIC, and WAIC. The deviance term denotes $-2\log p(y \mid \hat{\theta}_{mle})$ for BIC and AIC (cf., Eqs 4 and 5), whereas it is defined by $-2lppd_{waic}$, and $-2lppd_{loo}$ for WAIC and LOO, respectively (cf., Eqs 9 and 12). The penalty term of BIC is $klog(n)$, in AIC is $2k$, and for WAIC is $p_{eff}$ (see Eq 10), whereas LOO requires no penalty term. Note that the lower value of deviance and penalty terms yields the lower value of information criterion implying higher model predictive power. Accordingly, the best epileptogenicity hypothesis has a Delta score of zero (shown in red), whereas an epileptogenicity hypothesis with a Delta score larger than 10 is very unlikely.

Fig 4A and 4B show the computed BIC and AIC, respectively, assuming a uniform prior on the excitability values. We recall that in order to calculate BIC and AIC, we require the maximum log-likelihood (MLL), which is independent of the prior distribution. To overcome this limitation in the epileptogenicity hypothesis testing, we specify a truncated uniform prior on the excitability parameter defined in the ranges $[a, \eta_c - \Delta\eta]$, $[\eta_c - \Delta\eta, \eta_c]$, and $[\eta_c, b]$ corresponding to HZ, PZ, and EZ hypotheses, respectively. Here, $\eta_c = -2.05$, $\Delta\eta = 1.0$, $a = -6.0$, $b = 0.0$ (see S3(A) Fig). As shown in Fig 4A and 4B, both BIC and AIC correctly favor the true hypothesis (EZ shown in red), however, there is still strong support for the alternative hypotheses ($\Delta BIC_j \leq 4$, $\Delta AIC_j \leq 4$ for $j \in$ {PZ, HZ}). Note that if we do not truncate the prior on excitability parameter to compute the maximum likelihood estimate $\hat{\theta}_{mle}$, BIC and AIC are not able to distinguish different epileptogenicity hypothesis since the deviance and penalty terms in both BIC and AIC are identical for each of epileptogenicity hypothesis (the number of data and model parameters are identical for different epileptogenicity hypotheses). Moreover, from comparing Fig 4A and 4B, it can be seen that BIC pays the larger deviance and penalty terms than AIC, which yields the larger value of BIC than AIC.

Fig 4C and 4D illustrate the computed WAIC and LOO for EZ, PZ, and HZ hypotheses, when an uninformative prior is placed on the excitability parameter, whereas Fig 4E and 4F show the corresponding results while using an informative prior. Here, the prior distribution on the excitability parameter of the selected brain region was considered as a Gaussian distribution $\mathcal{N}(\mu_{hypo}, \sigma^2)$ with $\mu_{ez} = -1.6$, $\mu_{pz} = -2.4$, and $\mu_{hz} = -3.6$ corresponding to EZ, PZ, and HZ hypotheses, respectively. The parameter $\sigma$ was set to $\sigma = 0.01$ and $\sigma = 100$ to specify the informative and uninformative priors on the excitability parameter, respectively (see S3(B) and S3(D))). Note that the prior on the excitability parameter of all other brain regions is identical as $\mathcal{N}(-2.5, 1.0)$.

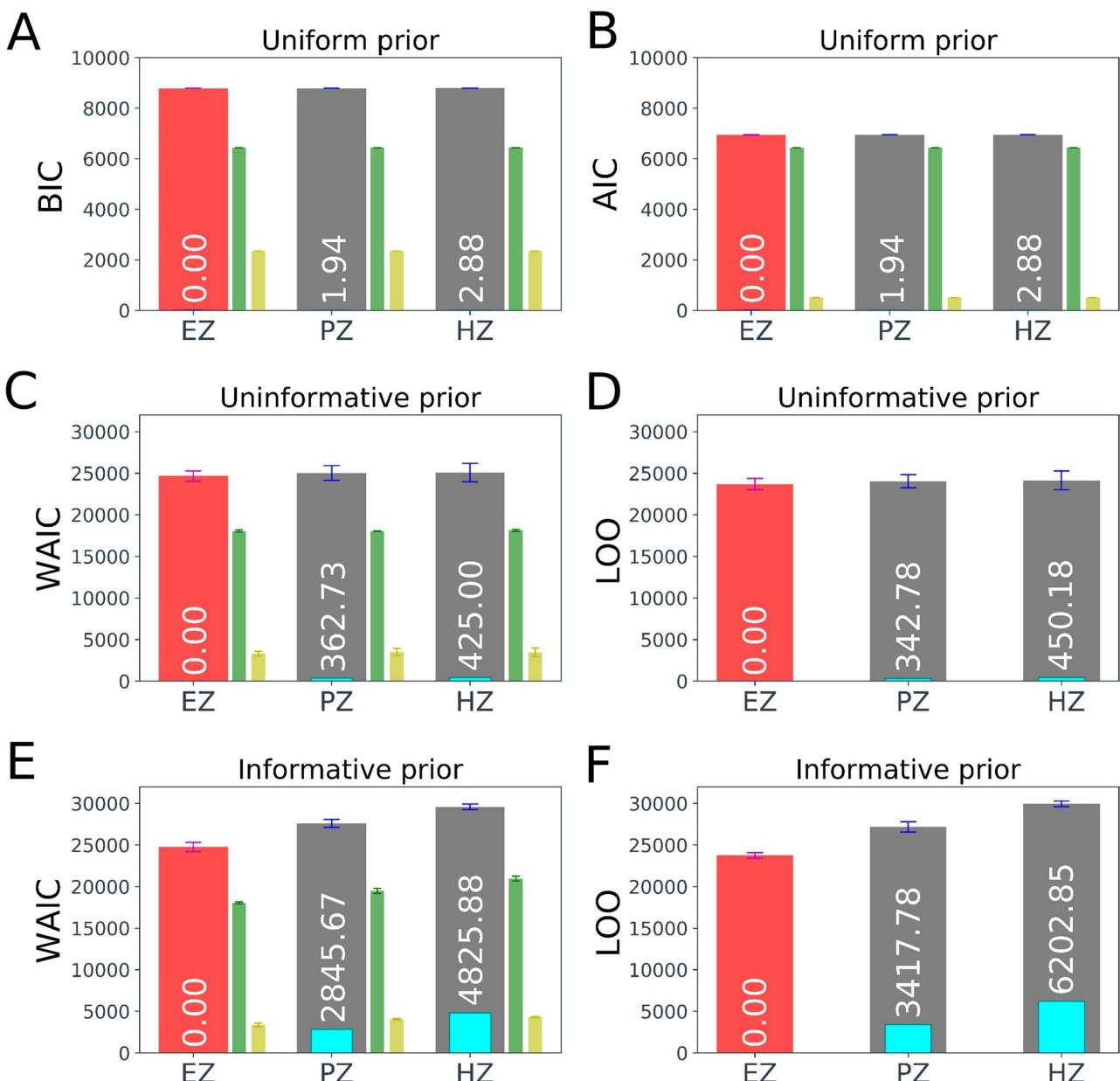

**Fig 4. Epileptogenicity hypothesis testing using classical and Bayesian information criteria.** The computed BIC, AIC, WAIC, LOO, and their Delta scores (color bars in cyan, text in white color) averaged over 4 estimations for each of the EZ, PZ, and HZ hypothesis, while the correct hypothesis for the selected brain node corresponds to the EZ (shown in red). The deviance (-2 times log predictive density, in green) and the penalty term (in yellow) are also displayed for BIC, AIC, and WAIC. (**A**) and (**B**) illustrate the computed BIC and AIC, respectively, assuming a truncated uniform prior on the excitability parameter. (**C**) and (**D**) show the computed WAIC and LOO, respectively, assuming an uninformative Gaussian prior on the excitability parameter ($\sigma = 100$). (**E**) and (**F**) show WAIC and LOO, respectively, but obtained by placing an informative prior on the excitability parameter ($\sigma = 0.01$). The smaller the value of each information criterion, the better the model's predictive ability. The best epileptogenicity hypothesis (smallest BIC/ AIC/WAIC/LOO) has a Delta score of zero (shown in red), whereas an epileptogenicity hypothesis with a Delta score of larger than 10.0 has essentially no support. The error bar represents the standard error.

From Fig 4C–4F we observe that when the prior on excitability parameter is uninformative, the true hypothesis (EZ shown in red) is correctly favored by both of WAIC and LOO, but with a large amount of uncertainty (see the standard deviations in Table 1). In contrast, when the prior is informative, both WAIC and LOO provide substantial support in favor of the

**Table 1. Summary of epileptogenicity hypothesis testing using the classical and Bayesian information criteria (BIC/AIC/WAIC/LOO) in terms of deviance (-2 times log predictive density), penalty term, and Delta scores.** Here, three hypotheses as EZ, PZ, and HZ are evaluated on the excitability parameter of a selected brain region, where the ground truth is EZ. The smaller deviance and penalty terms, the smaller the value of information criterion, the better the model's predictive ability. The correct hypothesis has a Delta score of zero.

| | | Hypothesis | | |
| --- | --- | --- | --- | --- |
| | | **EZ** | **PZ** | **HZ** |
| BIC | $-2\log p(y \mid \hat{\theta}_{mle})$ | 6435.83± 2.08 | 6437.77 ± 1.48 | 6438.70 ± 2.42 |
| | k (number of model parameters) | 255 | 255 | 255 |
| | n (number of data points) | 10080 | 10080 | 10080 |
| | $BIC_i = -2\log p(y \mid \hat{\theta}_{mle}) + k\log(n)$ | 8786.49 ± 2.08 | 8788.44 ± 1.48 | 8789.37 ± 2.42 |
| | $\Delta BIC_i = BIC_i - BIC_{min}$ | 0.0 | 1.94 | 2.88 |
| AIC | $-2\log p(y \mid \hat{\theta}_{mle})$ | 6435.83± 2.08 | 6437.774 ± 1.48 | 6438.70 ± 2.42 |
| | k (number of model parameters) | 255 | 255 | 255 |
| | $AIC_i = -2\log p(y \mid \hat{\theta}_{mle}) + 2k$ | 6945.83 ± 2.08 | 6947.77± 1.48 | 6948.70 ± 2.42 |
| | $\Delta AIC_i = AIC_i - AIC_{min}$ | 0.0 | 1.94 | 2.88 |
| WAIC (Uninformative prior) | $-2lppd_{waic}$ | 18053.88 ± 124.71 | 18071.15 ± 29.49 | 18140.93± 125.51 |
| | $p_{eff}$ | 3309.14 ± 277.61 | 3481.87 ± 431.05 | 3478.11 ± 501.51 |
| | $WAIC_i = 2(lppd_{waic} - p_{eff})$ | 24672.17 ± 614.68 | 25034.89 ± 890.64 | 25097.17 ± 1109.20 |
| | $\Delta WAIC_i = WAIC_i - WAIC_{min}$ | 0.0 | 362.73 | 425.00 |
| LOO (Uninformative prior) | $-2lppd_{loo}$ | 23693.80 ± 674.08 | 24036.58± 778.03 | 24143.98 ± 1136.79 |
| | $LOO_i = -2lppd_{loo}$ | 23693.80 ± 674.08 | 24036.58± 778.03 | 24143.98 ± 1136.79 |
| | $\Delta LOO_i = LOO_i - LOO_{min}$ | 0.0 | 342.78 | 450.18 |
| WAIC (Informative prior) | $-2lppd_{waic}$ | 18051.38 ± 108.21 | 19477.37 ± 297.94 | 20958.32 ± 289.23 |
| | $p_{eff}$ | 3353.22 ± 236.12 | 4063.06 ± 86.26 | 4312.69 ± 64.99 |
| | $WAIC_i = -2(lppd_{waic} - p_{eff})$ | 24757.83 ± 557.47 | 27603.50 ± 461.54 | 29583.71 ± 329.20 |
| | $\Delta WAIC_i = WAIC_i - WAIC_{min}$ | 0.0 | 2845.67 | 4825.88 |
| LOO (Informative prior) | $-2lppd_{loo}$ | 23759.25 ± 332.94 | 27177.02 ± 609.29 | 29962.104 ± 343.11 |
| | $LOO_i = -2lppd_{loo}$ | 23759.25 ± 332.94 | 27177.02 ± 609.29 | 29962.104 ± 343.11 |
| | $\Delta LOO_i = LOO_i - LOO_{min}$ | 0.0 | 3417.78 | 6202.85 |

correct EZ hypothesis, while the other epileptogenicity hypotheses have essentially no support ($\Delta WAIC_j \gg 10$, $\Delta LOO_j \gg 10$ for $j \in$ {PZ, HZ}). This is due to the larger deviance and penalty terms (shown in green and yellow, respectively) that are imposed by informative prior. Moreover, it can be seen that WAIC closely approximates LOO for each of epileptogenicity hypothesis. These results are reported in Table 1. According to these results, while using WAIC and LOO for epileptogenicity hypothesis testing, there is a decisive evidence in favor of the true epileptogenicity hypothesis if an informative prior is placed on the excitability parameter of the selected brain region.

## The effect of prior on the global coupling parameter

To demonstrate how the model performance is affected by the parameters of a Gaussian prior placed on the global scaling parameter $K$ (cf. Eq 2), here, the model accuracy in excitability parameter estimation and LOO cross-validation are computed as a function of the standard deviation in prior information. More precisely, the prior placed on the global scaling parameter $K$ is assumed as a Gaussian distribution $\mathcal{N}(\mu, \sigma^2)$, in which $\mu$ is fixed as the ground-truth value used in the simulation ($K = 1$), whereas $\sigma$ representing the uncertainty about the estimation varies from 0.01 to 1000. Fig 5A–5D show the changes in classification accuracy in EZ/PZ prediction, the z-scores in estimated excitability parameters (cf. Eq 14), the potential scale reduction

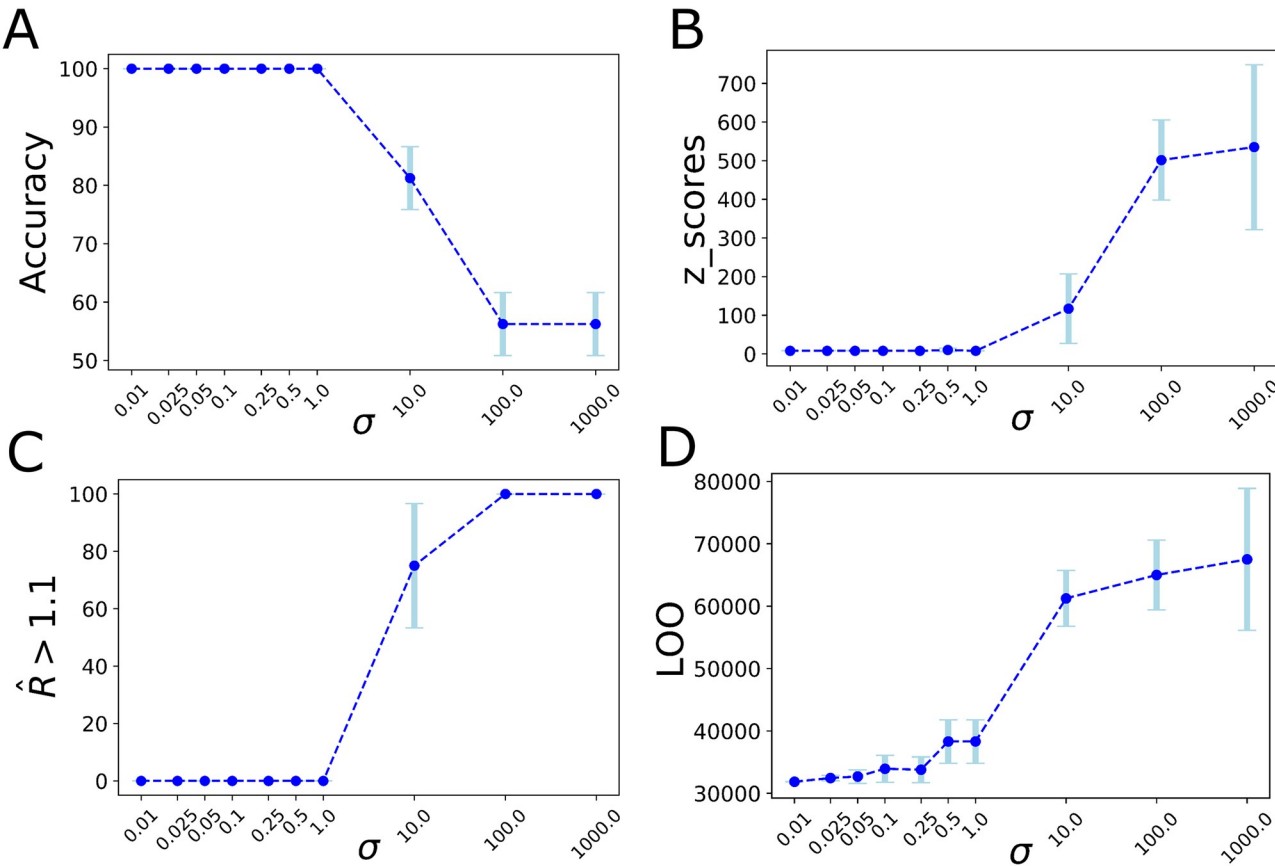

**Fig 5. Effect of the prior on the model performance in the epileptogenicity estimation.** Here, the prior placed on global coupling parameter $K$ is assumed as a Gaussian distribution $\mathcal{N}(\mu = 1, \sigma^2)$, where $\sigma$ represents the uncertainty about the estimation varying from 0.01 to 1000. (**A**) The classification accuracy in EZ/PZ prediction. (**B**) The z-scores in excitability parameter estimation. (**C**) The potential scale reduction factor $\hat{R}$ to monitor the HMC chains convergence. (**D**) LOO cross-validation as the out-of-sample model predictive accuracy. The figure plots the averages over 4 HMC chains. For $\sigma \leq 1.0$ all the HMC chains converged; for $\sigma = 10.0$ one out of 4 HMC chains converged; for $\sigma \geq 100.0$ all the chains failed to converge.

factor $\hat{R}$ for monitoring the HMC convergence, and LOO cross-validation as the out-of-sample model predictive accuracy, respectively, when the level of information in prior varies from significantly high to very low values. The figure plots the averages over 4 HMC chains randomly initialized in search space. From this figure, we observe that an informative or a weakly informative prior ($\sigma \leq 1.0$) is required to place on the parameter $K$ for achieving an accurate and reliable estimation in excitability parameters across brain regions (100% classification accuracy in EZ/PZ prediction, the z-scores close to zero, and the values of $\hat{R}$ lower than 1.1 for all of the hidden states and parameters implying that all the 4 HMC chains have converged). For $\sigma = 10$, only one out of 4 HMC chains converged, whereas for the uninformative prior ($\sigma \geq 100.0$) all the 4 chains failed to converge (classification accuracy close to zero, large values of z-scores, and $\hat{R}$ larger than 1.1 for all of the hidden states and parameters). Moreover, the estimated LOO values indicate that the higher level of information in prior (smaller $\sigma$) provides a higher level of predictive accuracy in epileptogenicity estimation across brain regions.

## Cross-validation of patient-specific connectome

In order to further explore the ability of fully Bayesian criteria in the evaluation of BVEP model components that vary across subjects (such as the patient's connectome), the simulated

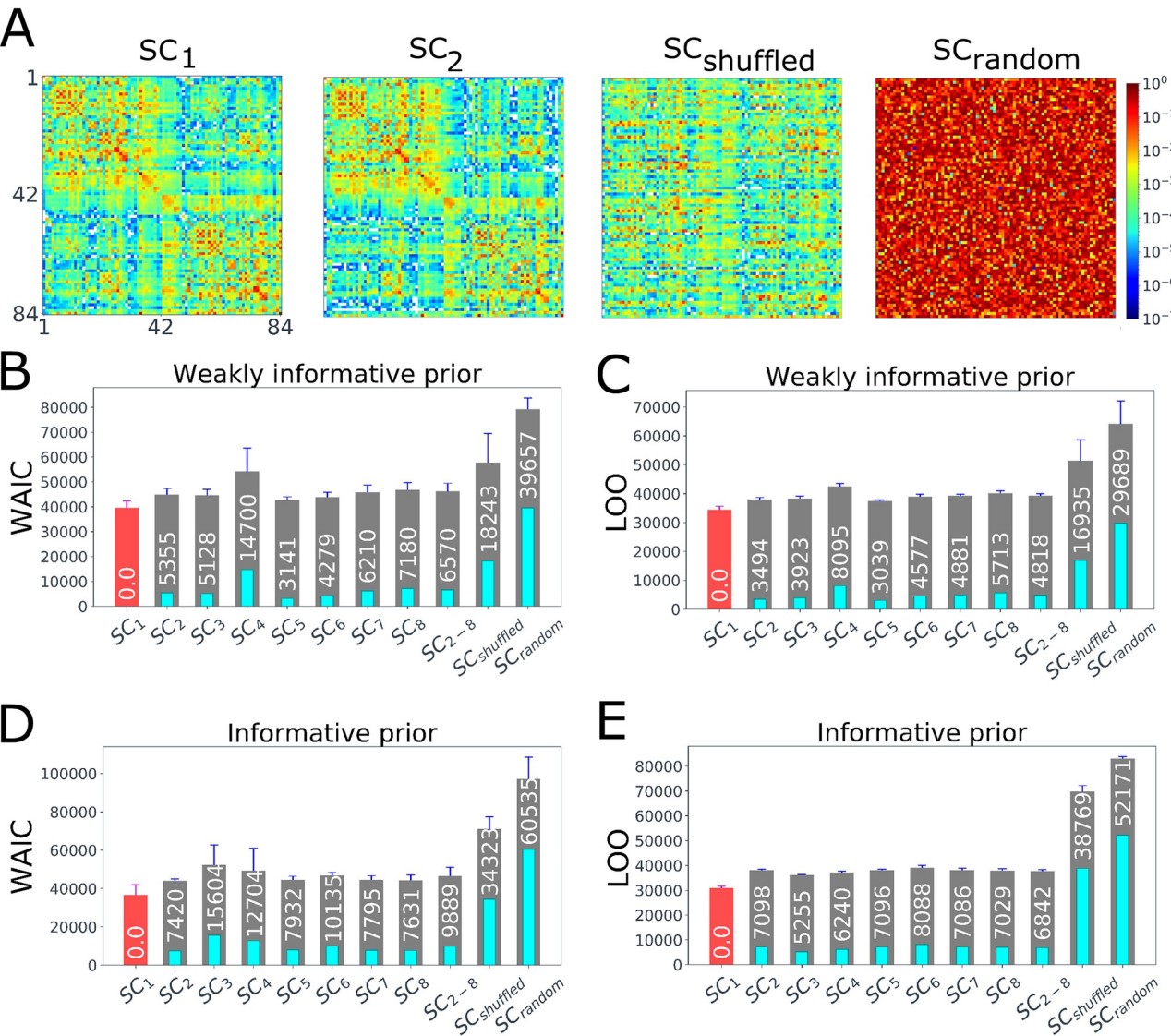

**Fig 6. Comparison of the BVEP model predictive accuracy by WAIC and LOO cross-validation for different SC matrices.** (**A**) Shown from left to right are: the patient-specific SC used in the simulation ($SC_1$), SC of a randomly selected patient ($SC_2$), shuffled SC of the patient ($SC_{shuffled}$), and a random connectivity matrix generated from a uniform distribution ($SC_{random}$). (**B**) and (**C**) show the computed WAIC and LOO, respectively, for different SC matrices by placing a weakly informative prior on the global scaling parameter. (**D**) and (**E**) illustrate WAIC and LOO, respectively, while using an informative prior distribution. The average of WAIC and LOO for other patients is also plotted (denoted by $SC_{2-8}$). The simulated data was generated by $SC_1$, and for each of the SC matrices, WAIC and LOO are averaged over 4 HMC chains (the Delta score for each of the SC matrices is shown in white text). The smaller the value of WAIC/LOO, the better the model's predictive ability (shown in red, with a Delta score of zero).

data generated through a patient's SC (see Fig 2) was fitted using different brain connectomes, then, WAIC and LOO cross-validation are compared for different SC matrices: SC of 8 randomly selected patients, a shuffled SC and a random connectivity matrix generated from a uniform distribution over [0.0, 1.0], across informative and weakly informative priors placed on the global scaling parameter $K$ (cf., Eq 2). Here, the spatial map of epileptogenicity is estimated for each of the SC matrices, while the prior distribution on the excitability parameter of all 84 brain regions included in the analysis is identical as $\mathcal{N}(-2.5, 1.0)$. Fig 6A illustrates the patient-specific SC used in the simulation (denoted by $SC_1$), SC of a randomly selected patient, shuffled SC, and a random matrix. The evaluated WAIC and LOO cross-validation for each of

the SC matrices while specifying a weakly informative prior distribution ($\sigma = 1.0$) over the global scaling parameter $K$ are shown in Fig 6B and 6C, respectively, whereas in Fig 6D and 6E an informative prior ($\sigma = 0.01$) is placed on the parameter $K$. The averages of WAIC and LOO for other patients are also plotted (denoted by $SC_{2-8}$). Fig 6B–6E show that in both cases of informative and weakly informative priors, WAIC and LOO cross-validation substantially favors the subject-specific SC (shown in red), whereas the alternative SC matrices have essentially no support ($\Delta WAIC \gg 10$ and $\Delta LOO \gg 10$ for the alternative SC matrices). However, a higher level of information in the priors provides more evidence for the subject-specific SC matrix. Comparing Fig 6B and 6D with Fig 6C and 6E, respectively, it can be seen that WAIC approximates LOO cross-validation for each of the SC matrices, but with larger deviations. Furthermore, we observe smaller Delta scores for random and shuffled SC matrices if a weakly informative prior is placed on the global scaling parameter compared to the informative prior (see Fig 6, middle row versus bottom row). This is due to the underestimation of parameter $K$ for shuffled and random SC matrices by the use of a weakly informative prior relative to the choice of an informative prior distribution. More precisely, in the case of placing an informative prior on $K$, this parameter is constrained around the ground-truth ($K = 1$) for all the SC matrices. In contrast, when a weakly informative prior is used, the estimated $K$ converges to zero for SC shuffle and SC random, in order to increase the model evidence against the fitted data. Nevertheless, using (weakly) informative prior on the global scaling parameter $K$, both WAIC and LOO cross-validation indicate that there is no support for the alternative SC matrices in comparison to the patient-specific SC matrix.

## Epileptogenicity hypothesis testing against empirical data

Finally, EZ prediction using fully Bayesian information criteria is tested for a patient with drug-resistant left temporal lobe epilepsy. To this end, the constructed BVEP model for this specific patient is fitted against the empirical SEEG recordings, and then three hypotheses as EZ, PZ, and HZ are compared to the clinical hypothesis of EZ (left Amygdala). Here, raw SEEG data is preprocessed to extract SEEG log power. Preprocessing involves high pass filtering from 10 Hz, computing the power over a sliding window, applying a log transformation, and then a low pass filter for data smoothing. Fig 7A shows the fitted BVEP model (in red), having very good agreement with the SEEG log power (in blue) for two implanted intracranial electrodes during the clinical evaluation. Fig 7B illustrates LOO cross-validation and Delta score for EZ,

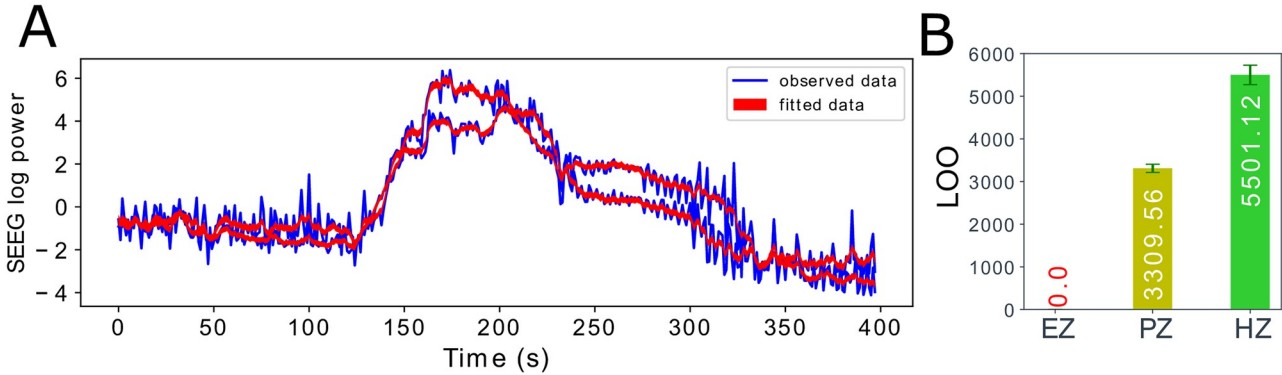

**Fig 7. Model fitting against empirical SEEG recordings and EZ hypothesis evaluation using fully Bayesian information criteria.** (**A**) SEEG log power of two implanted intracranial electrodes (in blue) versus the fitted envelope from BVEP model inversion (in red). (**B**) LOO cross-validation for the three hypotheses testing about clinical hypothesis (left Amygdala) as EZ, PZ, and HZ and their Delta scores (white text). Delta score of zero indicates the best hypothesis (EZ), whereas the alternative hypotheses (PZ and HZ) have essentially no support.

PZ, and HZ hypotheses when an informative prior ($\sigma = 0.01$) is placed on the excitability parameter of the selected brain region (left Amygdala). The model prediction by fully Bayesian information criteria provides substantial support in favor of the EZ hypothesis (Delta score of zero), which is in agreement with post-surgical clinical evaluation of this patient as seizure free.

## Discussion

Due to the substantial improvements in personalized medical treatment strategies, the individualized large-scale brain network modeling has gained popularity over the recent years [13, 64]. Using the whole-brain modeling approach, patient-specific information such as anatomical connectivity obtained from non-invasive imaging techniques is combined with mean-field models of local neuronal dynamics to simulate the individual's spatiotemporal brain activity at the macroscopic scale [28, 63, 101]. Based on the personalized large-scale brain network modeling, a novel approach to brain interventions has been recently proposed to integrate patient-specific information such as brain connectivity, epileptogenic zone and MRI lesions [13, 34]. This approach relies on the fusion of structural data of individual patients, mathematical modeling of abnormal brain activity, and mapping between source and sensor signals to predict the localization and evolution of epileptic seizures [13]. However, finding a set of parameters that yields the best possible model fit to the recorded data is a challenging task due to the high dimensionality and the nonlinearity involved in the model equations. In our recent study, we proposed a novel probabilistic approach namely Bayesian Virtual Epileptic Patient (BVEP) to establish a link between the probabilistic programming languages (PPLs such as Stan/PyMC3; open-source statistical tools for flexible probabilistic machine learning) and the personalized brain network modeling (e.g., the VEP brain model generated by TVB [59]). Using PPLs along with high-performance computing enables systematic and efficient parameter inference to fit and validate the large-scale brain network models against the patient's data [4].

Considering a set of competing hypotheses for a model structure, we would need to evaluate the model's predictive power in order to determine the best hypothesis that balance model complexity and predictive accuracy. For instance, in the context of epilepsy, different EZ hypotheses can be confronted directly against the patient's empirical data. In the BVEP model, according to the dynamical system properties to trigger seizures, the brain regions are divided into three main types as EZ, PZ, and HZ. For each brain region, it is then required to evaluate three epileptogenicity hypotheses in order to determine the hypothetical area responsible for the origin and early organization of the epileptic seizures. In this study, by linking the PPLs and the personalized large-scale brain modeling, we proposed a probabilistic framework for epileptogenicity hypothesis evaluation by estimating pointwise out-of-sample prediction accuracy from the existing posterior samples (cf. Fig 1). Systematic detection of EZ in the BVEP brain model aids to refine the first clinical hypothesis regarding the location of epileptogenic zone before epilepsy surgery. The patient-specific strategy is key to the improvement of the accuracy of resections or thermal lesions designed for modulating large-scale epileptic networks.

In this study, we demonstrated that using the recently developed MCMC algorithm known as No-U-Turn Sampler (NUTS), the spatial map of epileptogenicity is accurately estimated across different brain regions (cf. Fig 2). Based on the synthetic data generated by TVB, we achieved a very close similarity between the simulated and the predicted seizure activity regarding the initiation, propagation, and termination (cf. Fig 2**B**). For all 84 brain regions included in the analysis, the ground truth of excitability parameter was under the support of posterior predictive distribution (cf. Fig 2**C**). The convergence diagnostics and posterior

behavior analysis validated the reliability and accuracy of the estimations. The concentration towards large posterior shrinkage along with concentration towards small z-scores confirmed the accuracy of Bayesian inversion (cf. Fig 2**D**), while the posterior behavior analysis verified the convergence of MCMC estimates (cf. Fig 2**E**). These results indicate that the spatial map of epileptogenicity across all brain regions included in the analysis is accurately and reliably estimated by the NUTS algorithm.

Among several methods proposed for model comparison, the information-based criteria and cross-validation are two rigorous and standard approaches for measuring the model predictive accuracy, and thus as a selection rule for the model comparison problems [2, 46, 48]. The information-based criteria such as BIC, AIC, and WAIC balance model complexity and predictive accuracy to determine the most likely among a set of candidate models [45, 46, 48]. Within brain-imaging settings, several previous studies have focused on BIC and AIC [23, 24], and the dependence on the prior information has been considered as a drawback for the model comparison. However, as we have shown in the case of epilepsy, the fully Bayesian criteria such as WAIC and LOO cross-validation can be highly beneficial to improve the methodological analyses in this context. The classical information criteria such as BIC and AIC are based on the maximum likelihood estimates as the accuracy term and a simple function of the number of data/parameters as the correction term. This approach imposes a limitation on EZ hypothesis testing, where the number of model parameters and data are similar across different candidate models (see S5 Fig). In contrast, WAIC and LOO are fully Bayesian for measuring the pointwise out-of-sample prediction accuracy. This advantage allows us to integrate our prior knowledge in the model selection problem to provide a decisive evidence regarding the confusing terms (true positive and true negative), as shown in this study. Using BIC and AIC, our results indicated that the lack of information about the excitability parameter of a selected brain region leads to strong support for the alternative epileptogenicity hypotheses (see Fig 4**A** and 4**B**), whereas using WAIC and LOO, a high-level of information about the excitability parameter as encoded in the prior distribution provides substantial support in favor of correct hypothesis (see Fig 4**C**–4**F**). It is important to note that the prior information is not relevant in computing predictive accuracy; rather it is relevant in estimating the parameters. However, a substantial change in the prior will affect the computations of the marginal likelihood [45]. An inappropriate choice of prior can lead to weak inferences and thus poor predictions, whereas a large amount of information encoded through the informative prior distribution can provide decisive evidence in favor of the correct hypothesis. In the case of EZ hypothesis testing, we showed that an informative prior on excitability parameter imposes larger deviance (-2 times log pointwise predictive density) and penalty (variance of log pointwise predictive density) terms in the alternative hypotheses than uninformative prior (see Table 1). This provides a decisive evidence in favor of the true EZ hypothesis, as indicated by Delta score. Furthermore, we illustrated that WAIC closely approximates LOO cross-validation in our simulated conditions (see Fig 3**A** versus Fig 3**B**, and Fig 4**C** and 4**E** versus Fig 4**D** and 4**F**, respectively). This is in agreement with previous studies showing that WAIC is based on the series expansion of LOO cross-validation, and asymptotically they are equal [45]. In summary, these results demonstrate that WAIC and LOO can be used as a selection rule for determining the most likely among a set of competing hypotheses regarding the location of EZ in the personalized whole-brain models of epilepsy spread.

The BVEP brain model relies on the non-invasive structural data of individuals derived from a patient's DTI. Combining individual's anatomical information with mean-field models of seizure dynamics, in this study, the seizure initiation was defined as an intrinsic brain region property, whereas the seizure propagation was characterized by complex spatiotemporal dynamics of large-scale brain network. In other words, we assume that the seizure originates

from a local network and recruits candidate brain region(s) strongly coupled to the pathological areas by perturbing their stable dynamics (if $K = 0$, then there is no seizure recruitment). Based on this assumption, the EZ dynamics depend only on the excitability parameter, thus the model performance in EZ estimation remains accurate by the changes in SC matrix, whereas the seizure propagation depends also on the network properties, thus the model performance in PZ estimation depends critically on the strength of connections to these nodes (see S4(B) Fig). In general, using connectome-based modeling approach, the model predictive power can then dramatically depend on the estimation of the global coupling parameter ($K$), which scale different region's influence in perturbing the emergent dynamics of a selected region through structural connectivity (SC). An overestimation of $K$ leads to misclassification of PZ as HZ, whereas an underestimation of coupling may yield to misclassification of PZ as EZ. We observed that for convergence of BVEP model, an informative or a weakly informative prior is required on the parameter $K$, whereas using an uninformative prior the HMC chains failed to converge (cf. Fig 5). Although we ran 4 randomly initialized HMC with 200 sampling and 200 warmup iterations, running a longer chain with pre-defined initialization can improve the HMC convergence. Other brain imaging modalities such as fMRI/MEG can provide more information about the excitability of brain regions [102, 103] or the global scaling parameter [104, 105]. However, fitting simultaneous brain recordings with different spatial and temporal resolutions is more challenging. In addition to the functional component of the BVEP model, we examined the ability of fully Bayesian criteria in evaluating the model's structural component that vary across subjects. Fig 6 demonstrated that WAIC and LOO are able to determine the subject-specificity of structural data among a set of candidates. In particular, we observed a substantial evidence in favor of the subject's SC used in the simulation, whereas the SC of a randomly selected patient has essentially no support. This finding paves the way for future studies aiming to elucidate the specificity of connectome fingerprinting from a large cohort of patients with epilepsy.

Several previous studies in whole-brain modeling have used correlation as a scoring function for point prediction in order to measure the similarity between empirical and simulated data of large-scale brain activities [106–108]. On the contrary, we used the fully Bayesian WAIC and LOO for estimating pointwise out-of-sample prediction accuracy to investigate whether the probabilistic predictions on new data are accurate. In Fig 6, we showed that the prior information about the global scaling parameter affects WAIC ad LOO (as a measure of model predictive accuracy), and the Delta score (as a measure of relative evidence for model comparison). Both WAIC and LOO correctly select the subject-specific structural connectivity matrix, even if the prior on the global coupling parameter is weakly informative (cf. Fig 6B–6E). However, higher level of information in the priors provides more evidence for the best hypothesis among a set of candidates. Using the classical approach of model selection, one may find a high correlation value or a low root mean square error between the fitted model and observed data, independent of the level of information in priors. In other words, the scoring function for point prediction such as root mean square error is not an appropriate measurement for the overall performance of large-scale brain network modeling. These results point out the importance of integrating prior information in measuring the model's predictive accuracy for a more reliable estimation of brain dynamics from large-scale brain simulations.

In this study, we focused on synthetic data to validate the technical reliability of our approach, the self-tuning sampling algorithms (NUTS), and more importantly, the fully Bayesian information criteria and systematic cross-validation for measuring the out-of-sample prediction accuracy. Showing the estimations for a patient cohort requires a detailed non-trivial comparison with the clinical evaluations and outcome after surgery, involving significant other work such as group analysis (in particular homogenization of cohort), which was not

aimed in this study. This remains to be critically investigated in future work. However, in order to demonstrate how the proposed approach can be applied to empirical data we have shown an illustrative example in Fig 7.

It is important to note that a detailed comparison of classical and Bayesian information criteria for different model comparison problems was beyond the scope of the present study. Our focus in this technical note was to compare the classical and Bayesian information criteria in the BVEP brain model of epilepsy spread, where the penalty term (number of observation/parameters) is equal for all potential candidates. In the BVEP model, each network node's capacity for triggering seizures depends on its connectivity and the excitability. Depending on the value of the excitability parameter which is the target of fitting, the Epileptor model placed at network nodes exhibits different stability regimes. According to the dynamical property of Epileptor, the activity at each brain node can be classified into three main groups as EZ (exhibiting unstable fixed point), PZ (close to saddle-node bifurcation), and HZ (exhibiting stable fixed point). Based on this approach, it is required to evaluate three epileptogenicity hypotheses to determine the epileptogenicity of the brain regions, while the model/data are the same for the possible hypotheses. In such cases, by integrating the prior information, fully Bayesian information criteria such as WAIC and LOO cross-validation can be highly beneficial to provide a decisive evidence in favor of the best hypothesis. Using this approach, the practical implications of EZ hypothesis testing in a large cohort of patients to outline the region of the brain which is more susceptible to seizures remain to be investigated in future work.

## Supporting information

**S1 Fig. The scaling behaviour of executing an increasing number of Markov chains on an increasing number of computing nodes (1–32) on the KNL Booster partition of the JURECA supercomputer.** (**A**) The total simulation time remains constant even when the number of Markov chains increases. (**B**) The speedup achieved increases linearly with the number of computing units.
(TIF)

**S2 Fig. Estimation of the spatial map of epileptogenicity across different brain regions in the BVEP model.** Violin plots of the estimated densities of excitability parameters for 84 brain regions in two different analyses. To simulate the seizure activity for a virtual patient, the excitability value was chosen as $\eta_{ez} = -1.6$ corresponding to regions in EZ, and $\eta_{pz} = -2.4$ for the regions in PZ, whereas all the other regions were defined as HZ with $\eta_{hz} = -3.6$. (**A**) Three brain regions are selected as part of EZ at the nodes $EZ_{idx} \in \{9, 13, 34\}$, and two regions as PZ at the nodes $PZ_{idx} \in \{6, 12\}$. (**B**) Two brain regions are selected as part of EZ at the nodes $EZ_{idx} \in \{7, 12, 36\}$, and two regions as PZ at the nodes $PZ_{idx} \in \{8, 13\}$. The ground truth values are displayed by the filled black circles.
(TIF)

**S3 Fig. Different type of prior distributions placed on the excitability parameter in the BVEP brain model.** (**A**) The uniform prior truncated in the ranges $[a, \eta_c - \Delta\eta]$, $[\eta_c - \Delta\eta, \eta_c]$, and $[\eta_c, b]$ corresponding to HZ, PZ, and EZ, respectively. Here, $\eta_c = -2.05$, $\Delta\eta = 1.0$, $a = -6.0$, $b = 0.0$. (**B**) Informative prior defined by $\mathcal{N}(\mu_{hypo}, \sigma^2)$, where $\mu_{ez} = -1.6$, $\mu_{pz} = -2.4$, and $\mu_{hz} = -3.6$ correspond to EZ, PZ, and HZ hypotheses, respectively, whereas $\sigma$ implies our information about the epileptogenicity hypothesis. Here, informative prior with $\sigma = 0.01$. (**C**) Weakly informative prior with $\sigma = 1.0$ (**D**) Uninformative prior with $\sigma = 1000$.
(TIF)

**S4 Fig. Estimation accuracy across different structural and functional components of the BVEP model.** (**A**) Excitability values used for simulation ($\eta_i^*$) versus the estimated values ($\bar{\eta}_i$) for a selected brain region (node number 6). As the node dynamics are varied by changing the excitability parameter (functional component), the model inversion by NUTS algorithm demonstrates an accurate and robust estimation by recovering the ground truth. Dashed red line represents a perfect fit. (**B**) The classification accuracy in EZ and PZ prediction as the value of structural connections (SC) to the selected brain region is decreased from 100% to 0% of value used in simulation (SC*), i.e. SC = $(1 - \epsilon)$SC*. As the dynamics in EZ depend only on the excitability parameter, the EZ prediction remains accurate irrespective of changes in SC (shown in red). However, the seizure propagation depends also on the network properties, thus the model performance in PZ prediction depends critically on the strength of connections to these nodes (shown in yellow).
(TIF)

**S5 Fig. Epileptogenicity hypothesis testing using the classical and fully Bayesian information criteria for 2D Epileptor model.** Here, the brain node with the highest excitability value corresponding to a region in EZ was analyzed, and each of the EZ, PZ, and HZ hypotheses was averaged over 100 estimations to determine whether the computed information criterion is statistically significant for the true hypothesis. Panels (**A**) and (**B**) show BIC and AIC, respectively, computed for three EZ, PZ, and HZ hypotheses, while the ground truth of excitability parameter $\eta$ in the simulation was -1.5 (i.e., a brain region corresponding to EZ). Placing a uniform prior on each hypothesis, both BIC and AIC are averaged over 100 MLL estimations. In both BIC and AIC, there is no significant difference (n.s.) between different hypotheses. (**C**) and (**D**) show the computed WAIC and LOO, respectively, for three EZ, PZ, and HZ hypotheses. Here, we have placed an informative prior on the excitability parameter $\eta$ by using Gaussian distribution $\mathcal{N}(\mu_{hypo}, \sigma^2)$, where $\sigma = 0.01$, and $\mu_{ez} = -1.5$, $\mu_{pz} = -2.5$, $\mu_{hz} = -3.5$, corresponding to EZ, PZ, and HZ hypotheses, respectively. Each bar plot shows the average over 100 HMC chains randomly initialized in search space. In contrary to classical information criteria, both WAIC and LOO correctly favor the true hypothesis with a high level of statistical significance (EZ hypothesis as shown in red, ****$p \leq 0.0001$). Moreover, it can be seen that WAIC closely approximates LOO cross-validation.
(TIF)

## Acknowledgments

The authors also gratefully acknowledge the computing time granted through JARA-HPC on the supercomputer JUWELS at Forschungszentrum Jülich.

## Author Contributions

**Conceptualization:** Meysam Hashemi, Marmaduke M. Woodman, Viktor K. Jirsa.

**Data curation:** Huifang Wang.

**Formal analysis:** Meysam Hashemi.

**Funding acquisition:** Maxime Guye, Fabrice Bartolomei, Viktor K. Jirsa.

**Investigation:** Meysam Hashemi.

**Methodology:** Meysam Hashemi, Anirudh N. Vattikonda, Viktor Sip, Marmaduke M. Woodman.

**Resources:** Maxime Guye, Fabrice Bartolomei.

**Software:** Meysam Hashemi, Sandra Diaz-Pier, Alexander Peyser, Marmaduke M. Woodman.

**Supervision:** Viktor K. Jirsa.

**Validation:** Meysam Hashemi.

**Visualization:** Meysam Hashemi.

**Writing – original draft:** Meysam Hashemi, Anirudh N. Vattikonda, Viktor Sip, Sandra Diaz-Pier, Alexander Peyser, Huifang Wang, Maxime Guye, Fabrice Bartolomei, Marmaduke M. Woodman, Viktor K. Jirsa.

**Writing – review & editing:** Meysam Hashemi, Anirudh N. Vattikonda, Viktor K. Jirsa.

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
