## [Decision Letter · Decision Letter 0]

16 Nov 2020

Dear Dr. Hashemi,

Thank you very much for submitting your manuscript "On the influence of prior information evaluated by fully Bayesian criteria in a personalized virtual brain model of epilepsy spread" for consideration at PLOS Computational Biology.

As with all papers reviewed by the journal, your manuscript was reviewed by members of the editorial board and by several independent reviewers. In light of the reviews (below this email), we would like to invite the resubmission of a significantly-revised version that takes into account the reviewers' comments.

Reviewer 2 in particular raised important and substantial concerns regarding the appropriateness of using a low dimensional approximation of the high dimensional model, and the minimal biological insight provided for the wide readership of PLoS Computational Biology. These concerns must be addressed in any revision.  

We cannot make any decision about publication until we have seen the revised manuscript and your response to the reviewers' comments. Your revised manuscript is also likely to be sent to reviewers for further evaluation.

Sincerely,

Peter Neal Taylor

Associate Editor

PLOS Computational Biology

Samuel Gershman

Deputy Editor

PLOS Computational Biology

Reviewer's Responses to Questions

**Comments to the Authors:**

Reviewer #1: The paper discusses the important aspect of using prior knowledge and data in a bayesian setting for quantitative analysis of presurgical investigations within epilepsy surgery. The main ideas of the paper are presented well and I find no concerns with the data analysis.

As minor issues,

The VEP includes the SC map and a 6D epileptor model at each node. The inversion is done as I understand it using a 2D epilpetor model. What is the reason to include the 6D model instead of the 2D model at each node as both models managed to create the envolope that is used for data inversion.

The inversion is done using a highly non-linear generative model. How brittle is this inversion scheme. It worked fine in the study but the inversion was done on partly tailored data as I understand it.

Reviewer #2: The paper builds on the authors’ prior efforts to develop dynamical systems model of large-scale brain network activity. Here, a specific application of these models is considered: the statistical testing of epileptogenic dynamics within specific brain regions. Using a ground truth model, the authors generate synthetic data of epileptic activity; then, a low-dimensional model is used to facilitate Bayesian inference of parameters associated with the disposition (epileptic or not) of different regions within the network. The main result pertains to the demonstration that fully Bayesian model selection and hypothesis testing criteria are more suitable to this inference problem versus popular techniques such as AIC or BIC.

Main:

- In the primary finding, the authors generate 'ground truth' data from a high-dimensional dynamical neural model (the ‘virtual brain’); then they use Bayesian inferential techniques in order to perform hypothesis testing regarding the parameterization of the model based on observed model output. This inference is performed using a low-dimensional approximation of the model. In this setting, it is shown that fully Bayesian model selection techniques are able to adjudicate key parameters pertaining to nodal excitability in the full model. The issue as I see it is that the low dimensional model is obtained via formal dimensionality reduction of the high dimensional model, so it is relatively unsurprising that a Bayesian inferential technique will return high-quality parameter estimates in this setting. If the argument is that this work sets the stage for deploying this method on actual human data, then what one wants is a mis-matched ground truth model, wherein both the within- and across-node dynamics may deviate from those of the inference model. In other words, how would this inference technique perform if the "actual" brain deviated in its dynamics from those nominally embedded in the inference model?

- A key aspect of this pertains to assumptions regarding how structure/diffusion-based anatomical characterizations inform signal propagation throughout the network. Because these assumptions are essentially equivalent in the high-dimensional and low-dimensional model variants, it is entirely expected that predictions regarding signal transmission between nodes would be relatively consistent between the two. This also pertains to the results of Figure 5, which establishes that an individual’s structural connectivity parameterizing the low-dimensional model is best among a set of alternatives. At the point that the low-dimensional approximation is shown to be valid, I’m not sure what this added layer of Bayesian inference adds.

- Likewise, the demonstration of hypothesis testing on the single patient is conceptually problematic. It seems to me that the 'fitting' of the high dimensional model to the SEEG data is actually the key step here and this very poorly described in the paper (indeed, the entirety of the methods for this seem to be implied through a few lines in the results). If I understand it correctly, this fitting produces parameters in the high-dimensional model that reflect increased excitability in the Amygdala, which is then substantiated through the proposed hypothesis testing regimen. At the point that one has the high-dimensional model, then the hypothesis testing essentially amounts to the results of Figure 4. As such, I find this single-subject application to be quite superficial.

- At a high level, it is unclear to me that PloS Computational Biology is the right forum for this paper. The biological insight conferred by these results is minimal at best; the primary contribution is in the space of methodological enhancement. That the fully Bayesian or LOO methods outperform BIC or AIC is a reasonable and potentially useful finding, but it is also quite narrow in scope given above considerations. Indeed, as shown in Figure 4, BIC or AIC still "work" albeit with a lower level of statistical ambiguity.

- There are many grammatical issues throughout the paper that impede readability. A thorough round of copyediting is needed.

Minor:

- This is a stylistic qualm, but I feel that the first part of the Introduction is far too technical without providing any context as to the points about model selection, information criteria etc. If would be helpful to first discuss the models being considered, their goals and then move into the problem at hand. I had to read through to the results before I started understanding what the paper was trying to accomplish.

- Is there a reason the authors choose epileptic zone, as opposed to the clinical term of an epileptic focus?

- P8l102: probabilistic brain network approach modeling approach?

- Figure 4 and Table 1 seem quite redundant.

- The use of ‘virtual brain model’ in the abstract is awkward. A model is by definition ‘virtual’ insofar as it is not the actual system at hand. Adding ‘virtual’ seems superfluous.

Reviewer #3: In this paper Hashemi et al. provide a study of how different strategies for model comparison to infer node features in an epileptic network using the 'virtual epileptic patient' framework. The work represents a much needed step towards integrating empirical (clinical) data into modelling approaches and illustrates nicely many key issues and a few crucial insights. Whilst I think that it would benefit from addressing some major concerns, I belief overall the work represents an important contribution with valid insights.

Major

====

- The authors show a single example of two types of inference (one pertaining to node-identity, one pertaining to structural connectivity identity). Are these representative? Were these the first ones tried? Does the inference break down? If so, where?

- Underlying assumptions regarding ictogenesis in the epileptic brain: The approach presented here seems to make strong assumptions regarding the generation of seizures in the epileptic brain: As presented here seizures arise from the focally abnormal activity of individual nodes and connectivity only describes their subsequent spread. Seizures could therefore never be said to arise from the interaction of two (individually not 'EZ' nodes)? Is this an assumption the authors make? Could they clarify this in the text?

- There is clearly the potential for integration of this work into the clinic, and the desire of the authors to illustrate this potential. For this it would be helpful which hypothesis the approach could test, and which part of the presented analysis speak to these pre-defined types of hypotheses that need testing. How does e.g. the identification of patient specific S.C. matrices relate to clinical applicability? Is it just a validity demonstration? What validity does this show?

- The limitations of the model are not discussed sufficiently. Two apparently selective inference examples are shown without clear discussion as to where the limits of inference on this very high dimensional brain models would be. Examples of how then inference breaks down and where it is applicable would be useful.

Minor

====

- The manuscript would benefit from some (light touch) English language editing for readability. Furthermore there are several typos throughout the manuscript that could be corrected for the next submission ('self-tuninginvariant', 'Structrural brain network model', 'predict the envelop' etc.).

- Introduction: Why is epilepsy an appropriate model for the verification of this approach. Since the gold standard of empirical seizure progression is not easily knowable, why not choose to fine tune the inference strategies for these whole-brain models on more constrained empirical data?

- Regarding the model parameters: Fig 2E: Please elaborate on what the 200,000 model parameters are, how many of these are free to vary during model inference and how this apparently huge parameter space can be constrained by very limited datasets

- Regarding the S.C. reconstructions - there is a non-trivial link between structural connectivity (of any kind, but particularly of those measured with DTI-based imaging methods) and effective synaptic connectivity. Is this link mediated through a fitted parameter?

- More clinical context regarding the patient could be valuable if possible to include (particularly syndromic epilepsy diagnosis, final histological diagnosis after operation)

- Is the approach presented here sensitive to changes in Atlas resolution? How was the specific Atlas resolution utilised here chosen.

- for those not familiar with the tools, it might be worthwhile to introduce the relevant 'PPL' tools and cite their relevant references where available

- Are there examples of where the inference algorithms fail and do not produce the expected posterior shrinkage

- a substantial amount of the paper focusses on the interesting and relevant developments in sampling algorithms, measures for model comparison, and computing structure, which is only alluded to in the introduction

- for choice of EZ / PZ in the simulation, was existing connectivity between nodes taken into account or matched to empirical data somehow. I notice that these were spatially quite segregated in the model (but it is not clear whether they are in fact strongly connected or not). Is this representative of real patient data? I would have anticipated PZ and/or EZ nodes to be fairly well connected to each other and/or spatially contiguous

- different imaging modalities are frequently alluded to - I appreciate that this is relevant for the wider applications of TVB, but it is not clear how other imaging modalities (especially fMRI) will have a clear role in the near future for inference of ictal brain dynamics as presented here

Figure 2: Model fits vs Observation - indeed the fit seems very good and I am somewhat surprised that the fitted and the observed data 'start' seizures at the same time.

- is there any noise driving the system?

- what defines the point of seizure onset? Will every simulation produce a seizure at the same time (because of e.g. constant rate input current?). How are the initial states of the system defined (e.g. of the slow variables)? Are they fitted as well?

Figure 3: Colour bar axes would be helpful.

How do Epileptor model states relate to SEEG data? Is there additional conversion or are they taken to just represent the log of the raw SEEG signal? What montage is that signal calculated from? Does that have an impact on the inversion?

**Have all data underlying the figures and results presented in the manuscript been provided?**

Reviewer #1: Yes

Reviewer #2: Yes

Reviewer #3: **No: **- numerical results from simulation on which inference is built not available

- if anonymised patient data (e.g. SC maps, summary measures of seizure activity at individual nodes) is included as figures in this publication, this should also be made available as it already constitutes published data. If there are data protection concerns about publishing these anonymised summary data features, they also should not be represented in figures.

PLOS authors have the option to publish the peer review history of their article (what does this mean?). If published, this will include your full peer review and any attached files.

Reviewer #1: **Yes: **Gerald K. Cooray

Reviewer #2: No

Reviewer #3: No
---

## [Decision Letter · Decision Letter 1]

4 May 2021

Dear Dr. Hashemi,

Thank you very much for submitting your manuscript "On the influence of prior information evaluated by fully Bayesian criteria in a personalized virtual brain model of epilepsy spread" for consideration at PLOS Computational Biology. As with all papers reviewed by the journal, your manuscript was reviewed by members of the editorial board and by several independent reviewers. The reviewers appreciated the attention to an important topic. Based on the reviews, we are likely to accept this manuscript for publication, providing that you modify the manuscript according to the review recommendations.

Sincerely,

Peter Neal Taylor

Associate Editor

PLOS Computational Biology

Samuel Gershman

Deputy Editor

PLOS Computational Biology

[LINK]

Reviewer's Responses to Questions

**Comments to the Authors:**

Reviewer #1: I am happy with the changes to the manuscript.

Reviewer #2: I appreciate the authors genuine efforts to engage with my earlier comments and the revised manuscript is certainly improved in my view. My remaining comments are:

“We thank the reviewer for this comment, as the deployment on actual human data is indeed

our motivation. We do not agree though that the matched ground truth model is straight

forward to work. The Bayesian inversion of the reduced model is a very challenging task and

to the best of our knowledge, there is no similar high-dimensional problem with nonlinear

generative model in the literature of brain network modeling.”

I was not suggesting that the Bayesian inversion is an easy task. I recognize that even the 2D model involves many parameters and so the inference problem is not trivial at all. My only point was that the final outcome -- that the inference led to `correct' inference of excitability -- was expected given that the 2D and 6D models are essentially matched in their underlying dynamics. I generally feel that the authors revisions help make this point more transparent.

“Thank you for mentioning this point. In order to avoid the non-identifiability for fitting the

SEEG data, the contacts are selected according to the mean energy of bipolars to provide a

bijection map between source activity (generated by Epileptor) at brain regions and

measurement at electrodes. This point is now added in a subsection Stereotactic-EEG (SEEG)

data preprocessing in Material/Methods. In addition, the codes and documentation for SEEG

preprocessing is added in Github repository.”

I am still not convinced that this part of the paper has a point. The veracity of the Bayesian framework has already been established by this point in the paper, so the only reason to go through this would be to make a statement about the ability of the method to localize seizure foci in actual brains (there is no additional technical validation offered by this example). However, as the authors point out, this validation is beyond the scope of the paper. I think the authors need to be clear about what this example provides if it is not a validity study regarding the ability of the method to provide inferences that agree with clinical assessments.

“The expression “virtual brain” has become a commonly, albeit loosely, used terminology for

full brain modeling using connectomes and neural mass models at network nodes.

Historically, this derives from the use of the neuroinformatics platform The Virtual Brain

(https://www.thevirtualbrain.org)”

I must disagree that this is common terminology. I have no issue with the nickname “the virtual brain”. However, this is nonetheless a nickname and it is inappropriate for the authors to attempt to project this as a standard term for all dynamical models of brain networks. “virtual brain” should be used in quotes with citation and qualification as to what this term actually means.

Reviewer #3: The authors have addressed all my comments about the previous version of this manuscript.

**Have all data underlying the figures and results presented in the manuscript been provided?**

Reviewer #1: Yes

Reviewer #2: Yes

PLOS authors have the option to publish the peer review history of their article (what does this mean?). If published, this will include your full peer review and any attached files.

Reviewer #1: **Yes: **Gerald Cooray, M.D., Ph.D

Reviewer #2: No

Reviewer #3: No

**Have the authors made all data and (if applicable) computational code underlying the findings in their manuscript fully available?**

Reviewer #3: Yes

Figure Files:

Data Requirements:

Reproducibility:

References:

---

## [Editor Report · Decision Letter 2]

29 May 2021

Dear Dr. Hashemi,

We are pleased to inform you that your manuscript 'On the influence of prior information evaluated by fully Bayesian criteria in a personalized whole-brain model of epilepsy spread' has been provisionally accepted for publication in PLOS Computational Biology.

Best regards,

Peter Neal Taylor

Associate Editor

PLOS Computational Biology

Samuel Gershman

Deputy Editor

PLOS Computational Biology

---

## [Editor Report · Acceptance letter]

6 Jul 2021

PCOMPBIOL-D-20-01623R2 

On the influence of prior information evaluated by fully Bayesian criteria in a personalized whole-brain model of epilepsy spread

Dear Dr Hashemi,

I am pleased to inform you that your manuscript has been formally accepted for publication in PLOS Computational Biology. Your manuscript is now with our production department and you will be notified of the publication date in due course.

With kind regards,

Zita Barta
